# Structured Machine Theory of Mind from Agent Trajectories[*]

## Abstract

Predictive models of human behavior trained on large-scale trajectory data optimize for statistical accuracy without representing the mental states that causally generate behavior. Such models support prediction but not principled intervention: they cannot answer how an agent's behavior would change if its beliefs or preferences were different. We introduce Structured Machine Theory of Mind (SMToM), a framework that addresses this limitation by attributing explicit, independently supervised belief and desire representations from observed trajectories within a Belief-Desire-Intention (BDI) causal structure. The central architectural element is a goal head that consumes only the predicted mental-state channels and a current-trajectory embedding; counterfactual intervention on beliefs and desires is thus a direct operation. We instantiate SMToM on a controlled pedestrian navigation domain where ground-truth mental states are known by construction, enabling evaluation of attribution accuracy, local channel interventions, and trajectory-level counterfactual diagnostics. The resulting model, BDIBottleneck, improves over trajectory-only and unsupervised-context neural baselines. Moreover, it is competitive with a matched auxiliary-supervised context baseline while providing an editable mental-state interface that the unbottlenecked baseline lacks. Counterfactual experiments on desires confirm that substituting an agent's inferred preferences with a different activity type coherently shifts predicted destinations toward relevant locations. Counterfactual experiments on beliefs confirm that marking a location as unavailable in the agent's belief state reduces its predicted probability as a destination, though in a complex way. Together, these results demonstrate, in a controlled navigation setting, that explicit BDI-structured supervision is a viable foundation for interpretable and counterfactual analysis of longitudinal trajectory data.

## 1 Introduction

Longitudinal datasets of human behavior, including mobility traces, activity logs, and interaction records, are increasingly used to build predictive models of population dynamics. The dominant approach is to train a deep model end-to-end on behavioral sequences and optimize for predictive accuracy on held-out data. These models can capture rich statistical regularities across individuals, but they do so without representing the mental states that actually generate behavior. The consequence is a structural limitation: a model that does not represent beliefs and desires cannot answer causal questions. It can predict that agents with certain historical patterns tend to visit certain locations, but it cannot say what those agents would do if their beliefs about the environment changed, or how a shift in underlying preferences would propagate to observable behavior. In other words, the model supports prediction but not interpretation or planning.

The appropriate framing for this problem is Theory of Mind (ToM), the capacity to attribute beliefs, desires, intentions, and other mental states to explain observed behavior (Premack & Woodruff, 1978; Wimmer & Perner, 1983; Leslie, 1987). Human behavior is causally generated by mental states: an agent goes somewhere because they want something and believe it is available. A model that recovers those components from trajectories does not merely correlate histories with futures; it maintains a causal representation of the agent that supports principled intervention. Formally, this is the problem of Machine Theory of Mind (MToM)

---

[*]All ideas and experiments are from the authors. LLMs were used for editing the writing to improve clarity and conciseness. Code and data will be released upon acceptance.

(Rabinowitz et al., 2018): building systems that can infer latent mental states from observable behavioral traces. The core difficulty is that mental states are unobservable and the inference is ill-posed: any finite behavioral trace is consistent with many mental state configurations, and no ground-truth labels exist in the wild. A model must simultaneously specify the space of mental states and learn their relationship to behavior.

Prior work has addressed this through two paradigms that are directly comparable in our trajectory-based setting. Model-based approaches, exemplified by Baker et al. (2011), formalize attribution as Bayesian inverse planning: observed actions are explained by the beliefs and desires that make them rational. These methods are interpretable and support counterfactual reasoning by construction, but they require hand-specified dynamics and likelihood functions that rarely hold outside controlled settings. Learning-based approaches, exemplified by ToMnet (Rabinowitz et al., 2018), treat MToM as a meta-learning problem and infer mental states from behavioral context without hand-crafted models. The gain in flexibility comes at the cost of transparency: learned representations are opaque embeddings with no guaranteed correspondence to specific mental state components, and direct intervention on beliefs or desires is not a native operation.

We introduce *Structured Machine Theory of Mind* (SMToM), a framework that combines the explicit representational structure of model-based methods with the data-driven flexibility of learning-based ones. SMToM dedicates separate neural components to belief and desire attribution and routes their outputs through a structured bottleneck into goal prediction, following the Belief-Desire-Intention (BDI) framework (Bratman, 1987; Rao & Georgeff, 1995). Because beliefs and desires are explicit, independently addressable variables, counterfactual reasoning is a native operation. The model supports augmenting an agent's inferred beliefs or preferences directly and observing the resulting change in predicted behavior: for instance, what if this agent believed a particular location was closed? What would they likely do instead, given their historical behavioral patterns and subject to the causal BDI structure? We instantiate SMToM on a controlled pedestrian navigation domain where ground-truth mental states are known by construction, enabling rigorous evaluation of both attribution accuracy and counterfactual validity. Real-world behavioral datasets contain no ground-truth belief or desire labels; without such labels, a model's claim to infer mental states cannot be distinguished from learning a behavioral correlate, making controlled simulation the necessary starting point before extending to richer environments.

## 2 Background

Machine Theory of Mind (MToM) sits at the intersection of three literature streams: model-based Bayesian inverse planning, learning-based neural inference, and recent LLM-centered ToM reasoning. Model-based approaches cast attribution as inverse planning: from observed actions, infer latent beliefs and desires that make behavior rational. Classic Bayesian formulations infer latent goals from behavior (Baker et al., 2011), and later work extends this to joint inference over beliefs, desires, and percepts (Baker et al., 2017). Related links to inverse reinforcement learning, including maximum-entropy formulations of trajectory-based intent inference (Ziebart et al., 2008), have clarified when these inference procedures are identifiable and tractable in structured settings (Jara-Ettinger, 2019; Wu & Schrater, 2018). Their main strength is semantic clarity: beliefs and desires are explicit variables, so counterfactual intervention is principled. Their main weakness is model dependence: practical use requires hand-specified dynamics, utility structure, and likelihood models, and performance can degrade when real behavior violates those assumptions.

Learning-based MToM, introduced in ToMnet (Rabinowitz et al., 2018), replaces hand-crafted inverse models with neural predictors trained directly from behavioral data and demonstrates cross-agent generalization. Subsequent work extends this paradigm to agents with dynamic latent trait representations, improving cross-agent generalization (Nguyen et al., 2022). This paradigm is data-scalable, but intermediate representations are often opaque: latent embeddings can contain belief-like and desire-like factors without separating them into independently controllable channels. Interpretability therefore becomes post hoc, and direct intervention on specific mental-state components is not native to the architecture. Recent structured-output variants begin to close this gap by explicitly supervising intermediate mental-state variables. For example, explicit supervision of belief/intention channels improves both interpretability and predictive accuracy (Oguntola et al., 2021), and explicit belief prediction has been demonstrated in multimodal human interaction settings (Bortoletto et al., 2024).

LLM-based ToM extends the field through language-native reasoning: early reports of competitive false-belief performance (Kosinski, 2024; Strachan et al., 2024) motivated pipeline and agent-style extensions (Wilf et al., 2024; Zhao et al., 2025) and multimodal benchmark evaluations (Jin et al., 2024), though subsequent work reveals fragility under perturbation (Ullman, 2023) and below-human benchmark accuracy (Sap et al., 2022; Chen et al., 2024; Wu et al., 2023; Kim et al., 2023; Shapira et al., 2023). Our setting is methodologically disjoint: inputs are trajectories rather than text, outputs are explicit belief/desire probability vectors rather than language tokens, and the core evaluation requires surgical intervention on individual mental-state channels, an operation that is not natively supported by language model prompting alone. An LLM can generate a verbal prediction about an agent's likely destination, but it does not expose an intervenable internal belief/desire state that can be modified and re-propagated through a goal head; in principle, one could engineer such a proxy state with additional scaffolding, but that is outside the model's default training objective and would require substantial system design. The counterfactual capability demonstrated in Section 6 is therefore architectural rather than prompt-driven.

Recent work has explored hierarchical latent variable models as an alternative route to scalable mental-state attribution (Doering et al., 2026); the present work takes a distinct approach through explicit BDI supervision and native counterfactual intervention.

SMToM is positioned to combine strengths across these lines: the explicit mental-state semantics and interventionability associated with model-based work, and the data-driven scalability of learning-based predictors, while remaining native to trajectory inputs rather than prompt-mediated textual reasoning.

## 3 Problem Formulation

### 3.1 Setting and Notation

We model navigation on a graph-structured environment $G = (\mathcal{V}, \mathcal{E})$ with $M = 24$ points of interest (POIs), $\mathcal{V}_{\text{POI}} = \{v_1, \ldots, v_{24}\}$, grouped into $C = 6$ desire categories (four POIs per category). Each agent has a fixed category preference distribution and fixed within-category preference distributions. We denote the corresponding flattened POI-level preference representation by $\pi \in [0, 1]^M$, where

$$\pi_j = P(d = \text{cat}(v_j)) \, P(v_j \mid d = \text{cat}(v_j)).$$

This $\pi$ is a stable agent-level representation (used as supervision), not a separate episode-level sampling mechanism. Each agent also has a binary belief vector $b \in \{0, 1\}^M$, where $b_j = 1$ indicates the agent believes POI $v_j$ is open. The environment is physically all-open; false beliefs are induced only through $b_j = 0$ entries. Belief configurations are indexed by $\mathcal{B} = \{B_0, \ldots, B_7, BH0, BH1\}$, where $BH0$ and $BH1$ are held out for belief-configuration generalization evaluation.

### 3.2 The BDI Generative Model

We adopt a BDI generative prior (Bratman, 1987; Rao & Georgeff, 1995): each episode first samples a desire category $d$, then samples a concrete goal POI within that category after masking believed-closed POIs and renormalizing over believed-open POIs in the selected category. Thus, generation is hierarchical (category $\rightarrow$ POI), rather than direct sampling from the 24-dimensional vector $\pi$. The resulting trajectory is $\tau = (v_0, \ldots, v_T)$, and the sampled goal is the operational intention variable. We use "goal" and "intention" interchangeably throughout; in the BDI sense, intention refers to a committed plan, which in this navigation domain reduces to the discrete destination POI the agent has committed to reaching. The key source of behavioral variation is the interaction of desire and belief: when a preferred POI is believed closed, probability mass is reallocated to belief-feasible alternatives, often within category. Detailed simulator assumptions and the explicit belief-feasible set definition are provided in Appendix D.

### 3.3 The Generative Model as a Structural Causal Model

The generative process above can be represented as a structural causal model (SCM) (Pearl, 2009). An agent is characterized by its stable preferences $\pi$ and belief configuration $b$; each episode is then generated by the

recursive assignments

$$d := h_d(\pi, \varepsilon_d), \qquad g := h_g(d, b, \varepsilon_g), \qquad \tau := h_\tau(g, \varepsilon_\tau),$$

with mutually independent exogenous noises, yielding the acyclic graph $\pi \to d \to g \leftarrow b$ and $g \to \tau$. Here $h_d$ samples a category from $\pi$, $h_g$ samples a goal within that category after masking believed-closed POIs and renormalizing over $b$, and $h_\tau$ generates a noisily near-shortest path to $g$ (full forms in Appendix E).

The framework is thus *structured* in two different ways. First, $\pi$ and $b$ are explicit nodes whose only shared descendant is the goal, so "intervention" and "counterfactual" carry their standard meaning: the experiments in Section 6 are $\mathrm{do}(\pi)$ and $\mathrm{do}(b)$ single-node queries. Second, the model of Section 4 is a structured amortized *inverse* of this SCM: the desire and belief heads approximate the posterior over the exogenous nodes ($\tau \mapsto \hat{\pi}, \hat{b}$), while the goal bottleneck re-implements the forward goal mechanism. Intervening on a recovered channel and re-propagating through the goal head thus evaluates the corresponding interventional distribution by construction. We *assume* this SCM and supervise its latent nodes with simulator ground truth; we therefore make no claim of unsupervised identifiability, in contrast with causal representation learning (Schölkopf et al., 2021), and instead validate faithful recovery in a controlled setting where true mental states are known.

### 3.4 The Attribution Problem

Given an observed trajectory $\tau$, the attribution task is to infer latent desire $\pi$, belief $b$, and goal $g$. The model outputs $\hat{\pi}$, $\hat{b}$, and $\hat{g}$. Training uses simulator supervision for all three variables; at inference time, only trajectories and same-agent historical context are available. Early trajectory prefixes are compatible with multiple latent explanations, so accurate attribution requires combining partial-path evidence with learned regularities over agent preference and belief-conditioned destination choice.

### 3.5 Assumptions

The most fundamental assumption is that behavior (can be represented as if it) is generated by the belief-desire-intention structure described in Section 3.3: an agent acts on a desire over activity categories and a belief over POI availability. If behavior is not BDI-structured, then the attribution targets are not well-defined. We further take the space of mental states to be fixed and known: $\pi$ ranges over $C = 6$ categories and $b \in \{0,1\}^M$ over $M = 24$ POIs. The present framework does not learn goals or beliefs outside these predefined sets. Use of open-ended or data-discovered mental-state spaces, for instance through EM-style or Chinese-restaurant-process discovery of new categories, is left to future work.

We also assume agents are approximately rational: given a goal, an agent follows a noisily near-shortest path to it (the $g \to \tau$ mechanism). Observed trajectories are thus informative about latent goals; strongly irrational or adversarial routing would sever the link between intention and behavior. For computational simplicity, we assume a static environment with fixed beliefs and a single goal per episode. False beliefs therefore live entirely in $b$ rather than in world dynamics. Dynamic world state and online belief updating are natural, straightforward extensions (see Section 7). We also assume the exogenous noises are mutually independent and no unmodeled common causes, which licenses treating $\mathrm{do}(\pi)$ and $\mathrm{do}(b)$ as clean single-node interventions.

Finally, we note that our evaluation involves *supervised* identification of the latent channels. We can validate that the model recovers beliefs and desires precisely because ground-truth mental states are available in the controlled setting, not because they are argued to be identifiable from behavior alone. Extending to settings without such access will require use of proxy-based validation, which we discuss as the main path to external validity (see Section 7).

## 4 Model

### 4.1 Input Construction for BDI Attribution

BDIBottleneck is trained on *meta-episodes* that provide three trajectory streams per sample: (1) a same-agent desire context, (2) a same-agent recent context, and (3) the current trajectory prefix. Coordinates are normalized and encoded as variable-length masked sequences. Unless otherwise stated, we use $K = 10$

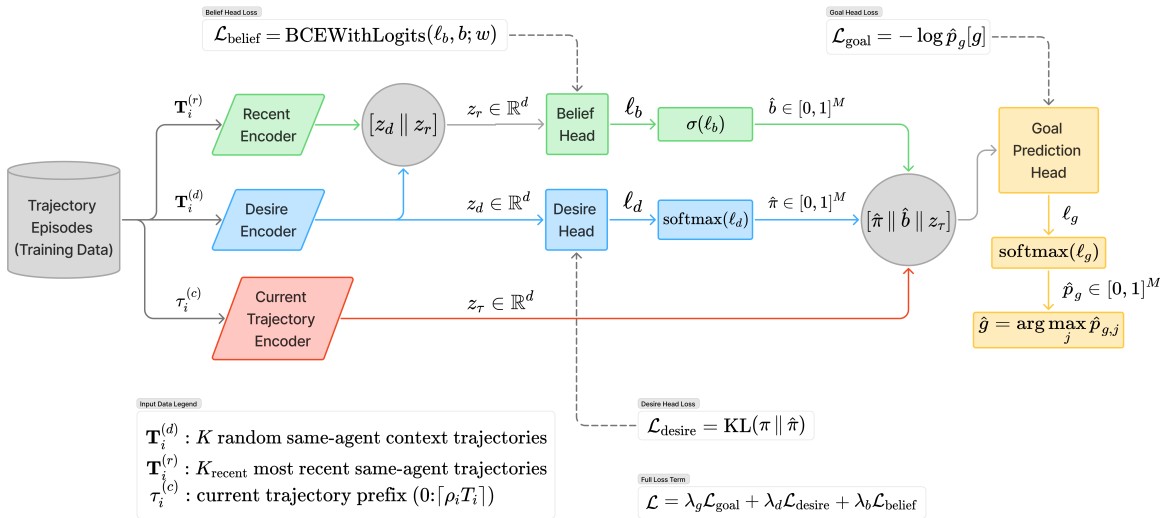

Figure 1: BDIBottleneck architecture. Three trajectory streams are encoded separately: random same-agent context for desire, recent same-agent context for belief comparison, and the current trajectory prefix for goal inference. The belief head uses $[z_d\|z_r]$, while the goal head uses the bottleneck representation $[\hat{\pi}\|\hat{b}\|z_\tau]$. Desire and belief channels can be intervened on directly for counterfactual analysis before goal prediction.

desire-context trajectories and $K_{\text{recent}} = 5$ recent trajectories. Supervision includes desire target $\pi$, belief target $b \in \{0,1\}^{24}$, and goal index $g \in \{1, \ldots, 24\}$.

## 4.2   BDI Bottleneck Architecture

The model uses three transformer encoders with shared input design but separate parameters: desire, recent, and current-trajectory encoders. Each sequence is projected from $\mathbb{R}^2$ to $\mathbb{R}^d$, position-encoded, and processed by a pre-LN stack. Figure 1 summarizes the full dataflow, including context construction, encoder outputs, head transformations, bottleneck concatenation, and the training losses.

Let $z_d \in \mathbb{R}^d$ be the pooled embedding of desire context, $z_r \in \mathbb{R}^d$ of recent context, and $z_\tau \in \mathbb{R}^d$ of the current trajectory. The intermediate heads are

$$\ell_d = W_d z_d + c_d, \qquad \ell_b = W_b[z_d\|z_r] + c_b, \tag{1}$$

with desire probabilities $\hat{\pi} = \text{softmax}(\ell_d)$ and belief probabilities $\hat{b} = \sigma(\ell_b)$.

The belief head uses $[z_d\|z_r]$ because the contrast between stable preference patterns and recent behavior is the primary cue for false-belief attribution: if an agent typically visits a POI (high signal in $z_d$) but has not done so recently (divergent $z_r$), this behavioral shift is evidence that the POI may be believed closed, paralleling the inference a human observer would make from the same behavioral pattern.

The defining architectural choice is the goal bottleneck: the goal head uses predicted mental-state channels rather than raw context embeddings,

$$\ell_g = \text{MLP}([\hat{\pi}\|\hat{b}\|z_\tau]), \tag{2}$$

which makes channel-level counterfactual intervention direct: desire and belief channels can be modified and re-propagated through the goal head without re-encoding context. Under our BDI interpretation, this goal variable is the operational intention variable.

### 4.3 Baselines and Comparison Models

We compare BDIBOTTLENECK against five models: two no-learning baselines (VISITFREQUENCY; BTOM (Informed)) and three neural models that isolate different sources of structure (GOALPREDICTOR; CONTEXTGOALPREDICTOR; AUXSUPERVISEDCONTEXTGOALPREDICTOR, abbreviated to AUXCONTEXT). VisitFrequency ranks POIs by goal frequency across the agent's $K$ same-agent desire-context episodes, capturing stable preference patterns independently of how much of the current trajectory has been revealed. BToM (Informed) applies Bayesian inverse planning with a Laplace-smoothed visit-frequency prior and a Boltzmann step-likelihood $P(v \mid u, g) \propto \exp(-\beta \cdot \mathrm{dist}(g, v))$ accumulated over the observed prefix, with $\beta$ tuned on the validation set; because the simulator generates near-shortest-path trajectories and the Boltzmann likelihood directly models step rationality under this process, BToM (Informed) approximates a Bayes-optimal predictor for this domain and serves as the approximate performance upper bound. GoalPredictor is a single-stream transformer over the current trajectory prefix only. ContextGoalPredictor uses the same three encoder streams as BDIBottleneck but removes explicit desire/belief supervision, predicting goal directly from concatenated encoder states $[z_d \| z_r \| z_\tau]$. AUXCONTEXT is the auxiliary-supervision control: it uses the same three encoders and the same auxiliary desire and belief heads/losses as BDIBOTTLENECK, but its goal head remains unbottlenecked,

$$\ell_g^{\mathrm{aux}} = \mathrm{MLP}_{\mathrm{aux}}([z_d \| z_r \| z_\tau]), \tag{3}$$

rather than consuming $[\hat{\pi} \| \hat{b} \| z_\tau]$. CONTEXTGOALPREDICTOR versus AUXCONTEXT measures the effect of adding desire/belief multi-task supervision to a context model; AUXCONTEXT versus BDIBOTTLENECK measures what is gained or lost when the same supervision is forced through an explicit BDI bottleneck.

### 4.4 Training Objective

Training minimizes a weighted multi-task objective:

$$\mathcal{L} = \lambda_g \mathcal{L}_{\mathrm{goal}} + \lambda_d \mathcal{L}_{\mathrm{desire}} + \lambda_b \mathcal{L}_{\mathrm{belief}}. \tag{4}$$

The goal term is standard cross-entropy, the desire term is KL divergence between $\pi$ and $\hat{\pi}$, and the belief term is class-weighted binary cross-entropy on $b$. We upweight belief supervision to compensate for false-belief sparsity (belief weighting $\lambda_b = 5.0$ for AUXCONTEXT and BDIBOTTLENECK). During training, only the current trajectory is randomly truncated to a sampled prefix; context streams remain intact. All four learned models share the same AdamW-based optimization setup and validation-goal model-selection rule, and each is trained with 3 independent random seeds; goal-inference and ablation results report the mean across seeds, with shaded bands in figures indicating $\pm 1$ standard deviation. Full per-term formulas, weighting details, and optimizer settings are provided in Appendix D and B.

BDIBOTTLENECK and AUXCONTEXT have matched auxiliary targets, weights, optimizer, context sizes, random seeds, and model-selection rule. The architectural difference is the goal-head input: BDIBOTTLENECK must predict through the explicit mental-state probabilities, whereas AUXCONTEXT may use whatever information is preserved in the raw context embeddings. This distinction is central for interpreting the results: raw goal accuracy reflects both useful auxiliary supervision and any cost of forcing the predictor through a low-dimensional, human-interpretable mental-state interface.

Although mental-state labels are available during training, they are absent at inference time. Models must recover beliefs and desires from trajectory context alone, exactly as a human observer would. The training labels identify targets for intermediate representations, but do not simplify the inference problem at test time. The counterfactual experiments in Section 6 thus provide an additional test of representational quality beyond attribution accuracy. If the intermediate channels were merely correlated with training labels rather than causally structured, surgical intervention on a single channel would not produce selective, directionally consistent shifts in goal predictions while holding the remaining channels fixed.

## 5 Experiments

Our empirical study evaluates whether the BDI bottleneck captures mental-state structure in a way that is both predictive and causally interpretable. We therefore emphasize two measures: (1) attribution performance under distribution shift across agents and belief configurations; (2) counterfactual validity via interventions on intermediate desire and belief representations while holding fixed remaining inputs to the goal head.

## 5.1 Data Splits

The simulator produces four disjoint episode splits. The `train` split contains episodes from 15 training agents under belief configurations $B_0$ through $B_7$ and is used for parameter learning. The `val` split uses the same agent family and configuration set, but remains disjoint at the episode level for model selection and early stopping. The `test` split is also episode-disjoint and measures in-distribution generalization for the same agent family. The `test_new_agent` split contains held-out agents whose category-level and within-category Dirichlet preference parameters are sampled independently from those of the training archetypes, producing agents with distinct preference profiles not seen during training, and tests whether learned representations transfer to unseen agents.

Training uses 2,040 episodes from 15 agents across 8 belief configurations ($B_0$–$B_7$); validation, test, and new-agent splits each contain 480 episodes from the same configurations. The two held-out belief splits (`test_bh`, `test_new_agent_bh`) contribute 480 episodes each under configurations `BH0` and `BH1` (full split sizes in Appendix Table 5).

Belief-configuration generalization is evaluated separately through `BH0` and `BH1`, which are never used for training and appear only at evaluation time in both `test` and `test_new_agent`. These held-out settings instantiate compound false-belief patterns that are structurally distinct from $B_0$ through $B_7$.

## 5.2 Generalization to New Agents

The primary attribution comparison is between `test` and `test_new_agent`. A model that overfits agent-specific trajectories should exhibit a large degradation on `test_new_agent`, whereas a model with agent-agnostic mental-state structure should remain stable.

For goal inference, we evaluate path fractions $\rho \in \{0.1, 0.2, \ldots, 1.0\}$, where only the first $\lceil \rho T \rceil$ nodes of the current trajectory are observed. In the main paper, we report top-1 accuracy; top-5 results are reported in Appendix I. This fraction sweep measures how quickly each model disambiguates destination intent as evidence accumulates.

Two additional diagnostic analyses are reported in Appendix F: a belief-head sensitivity analysis evaluating how well the belief head separates open and closed belief states at the (agent, POI) level as a function of training-visit exposure, and a distractor robustness evaluation measuring whether the model correctly ranks the true goal above a spatially proximate low-preference POI at the moment of closest approach.

## 5.3 Counterfactual Evaluation

Attribution accuracy alone does not establish modular causal representation. The latter requires that interventions on desire and belief channels induce coherent and selective changes in the predicted goal distribution. We distinguish two counterfactual type. First, we consider local channel interventions: the observed trajectory prefix remains fixed, one intermediate mental-state channel is edited, and the frozen goal head is re-evaluated. This tests whether the learned representation has an interventionable causal interface, but it does not show that the whole future trajectory would be (re)generated under the edited mental state. The second type, reported as an additional diagnostic, constructs trajectory-level counterfactuals after the intervention point and asks whether the model's predictions align with a new simulated or planned future.

For each evaluation episode $i$, we use inferred channels $(\hat{\pi}_i, \hat{b}_i)$ and a current-prefix trajectory embedding $z_{\tau,i}$, and obtain the goal distribution from the deterministic goal head $f_g(\cdot)$.

For desire counterfactuals, we intervene only on the desire channel while keeping belief and trajectory fixed:

$$\hat{p}_i^{\text{cf}} = f_g(\tilde{\pi}, \hat{b}_i, z_{\tau,i}). \tag{5}$$

The evaluation is run at path fractions $\rho \in \{0.25, 0.50, 0.75, 1.00\}$, where $z_{\tau,i}$ is re-encoded from only the first $\lceil \rho T_i \rceil$ nodes of the current trajectory. It is therefore not a full-trajectory counterfactual for the current path. We evaluate two prototype constructions for $\tilde{\pi}$: (i) a model-derived category prototype, computed as the mean inferred desire vector over episodes with sampled desire category $c$ (i.e., $\tilde{\pi}_c^{\text{model}} = |\mathcal{I}_c|^{-1} \sum_{i \in \mathcal{I}_c} \hat{\pi}_i$), and (ii) a handcrafted category-only prototype, uniform over POIs in category $c$ and zero elsewhere ($\tilde{\pi}_{c,j}^{\text{hand}} = 1/|\mathcal{V}(c)|$ for $v_j \in \mathcal{V}(c)$, else 0). For each receiver episode, we sweep all six category prototypes and measure the shift in category-level goal probability relative to the no-swap baseline, producing a $6 \times 6$ delta matrix per split and evaluated fraction.

Belief interventions follow the same fraction-sweep design as desire counterfactuals, with evaluation at $\rho \in \{0.25, 0.50, 0.75, 1.00\}$ on episodes whose realized goal is within the top-3 preference ranks for that episode. As in the desire case, $z_{\tau,i}$ is re-encoded from only the first $\lceil \rho T_i \rceil$ nodes rather than using the full current trajectory. We report the 50% ($\rho = 0.50$) results as the primary operating point in the main text, with the full fraction sweep in the appendix. Let $c(g)$ denote the category containing goal POI $g$. We define an intervened belief vector $\tilde{b}_i$ by setting the goal belief to closed and same-category alternatives to open:

$$\tilde{b}_{i,g} = 0, \quad \tilde{b}_{i,j} = 1 \; \forall j \in \mathcal{V}(c(g)) \setminus \{g\}, \quad \tilde{b}_{i,k} = \hat{b}_{i,k} \; \forall k \notin \mathcal{V}(c(g)), \tag{6}$$

with desire and trajectory inputs fixed. Let $\hat{p}_i = f_g(\hat{\pi}_i, \hat{b}_i, z_{\tau,i})$ and $\hat{p}_i^{\text{cf}} = f_g(\hat{\pi}_i, \tilde{b}_i, z_{\tau,i})$. We then compare original and intervened goal distributions using two paired outcomes: alternative gain, where $\Delta_{\text{alt}} = \sum_{j \in \mathcal{V}(c(g)) \setminus \{g\}} (\hat{p}_{i,j}^{\text{cf}} - \hat{p}_{i,j})$ and $\Delta_{\text{goal}} = \hat{p}_{i,g}^{\text{cf}} - \hat{p}_{i,g}$. Statistical significance is assessed with one-sided random-sign permutation tests on mean difference, one-sided exact sign tests, and percentile bootstrap confidence intervals.

The local interventions above isolate the goal head's response to channel edits, but does not show that those edits track a real agent's response. We test this external validity with a *model-planned* diagnostic that reuses the same belief and desire edits at a fixed intervention node (the end of the observed prefix). The frozen model predicts a counterfactual goal $g_{\text{cf}}$ through its edited channel; independently, the simulator, placed in the same edited mental state, resamples the goal $g_{\text{sim}}$ that its agent would in fact pursue and routes a future toward it from the intervention node. The question is then whether the model's counterfactual prediction agrees with the agent's actual counterfactual behavior, measured by agreement: top-1 ($g_{\text{cf}} = g_{\text{sim}}$) and top-5 ($g_{\text{sim}}$ within the edited model's top-5), tested against a within-group shuffled-goal permutation baseline. As a finer-grained supplement, we also report how much probability the edited model places on the original, model-selected, and simulator goals ($\Delta_{\text{orig}}, \Delta_{\text{sel}}, \Delta_{\text{sim}}, \Delta_{\text{sim}-\text{cf}}$, defined precisely and reported in full in Appendix F.8), assessed with random-sign permutation tests on seed-agent cluster means. This diagnostic does not exhaustively capture behavioral counterfactual identification, but does test whether an edited mental-state channel yields goal predictions consistent with the agent's counterfactual behavior under controlled simulator and routing assumptions.

## 6 Results

### 6.1 Goal Inference Performance

Figure 2 compares top-1 goal accuracy for the goal-inference models on in-distribution test and held-out new-agent episodes; Table 1 reports the same fair baseline under held-out belief configurations. We treat BToM (INFORMED) as an approximate upper bound: it combines a visit-frequency prior estimated from same-agent context with a Boltzmann likelihood that directly models the near-shortest-path generative process, but it does not use the agent's true preference prior. Its accuracy therefore reflects a strong, but approximate, limit on what trajectory observations can support in this domain. On the test split, BToM (INFORMED) reaches 34.8% at 50% reveal and 67.7% at 90% reveal (40.4% and 71.3% on new agents), rising steeply at longer reveals as accumulated Boltzmann likelihood concentrates on the true goal. Among the learned models, BDIBOTTLENECK is closest to this ceiling at nearly all path fractions, despite having no access to the true movement model or ground-truth priors. The fair AUXCONTEXT comparison qualifies this interpretation: auxiliary mental-state supervision itself improves a context model, while the bottleneck contributes an explicit intervenable interface rather than guaranteeing uniformly higher raw accuracy.

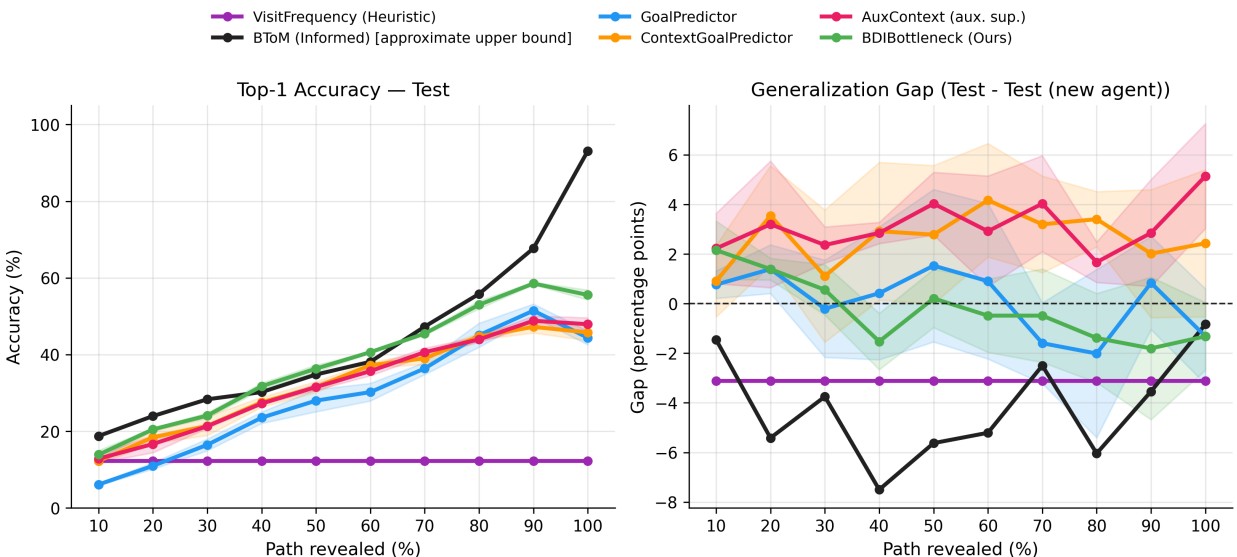

Figure 2: Top-1 goal inference including the matched auxiliary-supervised AuxContext baseline. Left: test-set accuracy (BToM (Informed) shown as approximate upper bound). Right: generalization gap ($\Delta = $ test $-$ new-agent) in percentage points; positive values indicate lower performance on unseen agents. Shaded bands: $\pm 1$ SD across 3 training seeds (learned models only).

BDIBottleneck reaches 55.56% on test and 56.88% on new agents at full reveal, and leads the trajectory-only and unauxiliary context neural baselines at 50% reveal (36.32% test, 36.11% new-agent), indicating stronger early disambiguation under partial observability. Several models exhibit a modest decline in accuracy from 90% to 100% reveal, but this is a pooling artifact: at full reveal the goal node is included in the trajectory input but its contribution is diluted by mean pooling over all preceding steps since there is no explicit termination signal. 90% reveal, where the agent is approaching but not yet at the goal, can thus be a more informative operating point for the pooling-based encoders. The wide gap between BDIBottleneck and BToM (Informed) at 100% reveal (55.56% vs. 93.1%, respectively) arises because the accumulated Boltzmann likelihood pins the unique destination consistent with a near-shortest-path route at full reveal, an advantage the neural models cannot match without the true movement model. At intermediate fractions, where the prior matters more relative to the accumulated likelihood, the learned contextual prior in BDIBottleneck closes much of this gap and marginally exceeds BToM (Informed) in the 40–60% interval (31.74% vs. 30.21% at 40%; 36.32% vs. 34.79% at 50%; 40.62% vs. 38.12% at 60%). Detailed 10%-increment values are reported in Appendix Table 20. VisitFrequency, which relies solely on a desire-context prior independent of trajectory length, achieves a constant 12.3% on test (15.4% on new agents); BDIBottleneck surpasses this from the earliest fractions on held-in agents and from 20% onwards on held-out new agents, confirming that trajectory evidence provides meaningful signal beyond the preference prior even at short reveals.

Comparison with ContextGoalPredictor shows the combined effect of mental-state supervision and the bottleneck: with matched encoder inputs and architecture, BDIBottleneck numerically leads at all path fractions on both standard splits and the advantage grows with reveal length. McNemar's exact tests with Holm–Bonferroni correction confirm that BDIBottleneck significantly outperforms GoalPredictor at most path fractions on both splits; gains over ContextGoalPredictor reach significance at longer reveals (80–100%) and on held-out agents (70–100%). The stricter AuxContext control shares significant structure with BDIBottleneck, so is analyzed against the held-out belief configurations below.

Table 1 shows the same evaluation under held-out belief configurations. This table should be read as a decomposition rather than a simple ranking. GoalPredictor tests trajectory evidence alone; ContextGoalPredictor adds same-agent context; AuxContext asks how much raw accuracy can be obtained

Table 1: Top-1 accuracy (%) at 25/50/75/100% path reveal for belief heldout splits. AuxContext is the unbottlenecked context model trained with the same auxiliary desire/belief losses as BDIBottleneck. Underline: BToM (Informed), approximate upper bound on achievable accuracy. **Bold**: best result per column among learned models.

| Model | Test (belief heldout) | | | | Test (new agent, belief heldout) | | | |
|---|---|---|---|---|---|---|---|---|
| | 25% | 50% | 75% | 100% | 25% | 50% | 75% | 100% |
| VisitFrequency | 13.33 | 13.33 | 13.33 | 13.33 | 14.17 | 14.17 | 14.17 | 14.17 |
| *BToM (Informed) [approximate upper bound]* | 24.79 | 33.96 | 48.33 | 94.38 | 29.79 | 39.79 | 58.33 | 94.17 |
| GoalPredictor | 12.50 | 25.83 | 40.42 | 42.50 | 10.35 | 27.50 | 41.04 | 43.82 |
| ContextGoalPredictor | 17.36 | 26.32 | 38.75 | 41.25 | 15.35 | 28.26 | 37.57 | 41.53 |
| AuxContext | 16.25 | 28.47 | 40.28 | 43.61 | **17.50** | 28.82 | 40.90 | 44.86 |
| BDIBottleneck (Ours) | **19.44** | **34.38** | **47.92** | **51.87** | 17.29 | **37.99** | **52.29** | **55.90** |

by adding the same auxiliary desire/belief losses without a bottleneck; and BDIBottleneck asks whether an explicit mental-state interface can retain competitive prediction while enabling direct interventions. Under the same 300-epoch paper budget, AuxContext improves over the unsupervised ContextGoalPredictor at most held-out belief fractions, confirming that auxiliary mental-state supervision is itself useful. BDIBottleneck nevertheless remains the best learned model in seven of the eight held-out belief columns: for example, at full reveal it reaches 51.87% on held-in belief-heldout episodes and 55.90% on held-out-agent belief-heldout episodes, compared with 43.61% and 44.86% for AuxContext. The only exception is the 25% held-out-agent belief-heldout column, where AuxContext is higher by 0.21 percentage points. The resulting evidence supports a calibrated conclusion: auxiliary supervision accounts for part of the raw goal-prediction improvement, while the BDI bottleneck can retain strong held-out-belief prediction and provides the structured channel on which the desire and belief counterfactual analyses below depend.

## 6.2 Attribution Quality of the Desire and Belief Heads

Figure 5 in Appendix B shows diagnostic training curves for the desire and belief heads across 300 epochs. The desire head converges to a validation KL divergence of 0.22 against the ground-truth preference distribution (from 0.69 at epoch 1), indicating stable and consistent desire attribution. For the belief head, the headline 76.1% overall accuracy on the 24-entry binary belief vector is dominated by the majority open-class entries; the false-belief F1, which measures precision and recall specifically on the 0–5 believed-closed entries per episode (0 for the all-open configuration, 3–5 for configurations with false beliefs), is the more informative metric. Measured on the validation set during training, false-belief F1 peaks at 40.4% in early training and converges to 33.3% at the selected checkpoint. An all-open predictor achieves 0% F1; the converged value therefore reflects genuine false-belief discrimination beyond trivial baselines. The declining F1 trajectory across training reflects the multi-task objective: as training progresses the cosine schedule increasingly concentrates gradient signal toward goal accuracy, and the belief head trades some false-belief sensitivity for compatibility with the goal loss. The functional test of causal structure in the converged belief representation is provided by the counterfactual experiments in the following sections. Additional diagnostic analyses—distractor robustness, belief-head sensitivity, sparse-versus-dense belief evidence, channel-use diagnostics, robustness sweeps, and trajectory-level counterfactual checks—are found in Appendix F with full implementation details and results.

Finally, to isolate each mental-state channel's contribution, we trained two ablations of BDIBottleneck: NoDesire (desire encoder and head removed; goal conditioned on belief and trajectory only) and NoBelief (recent encoder and belief head removed; goal conditioned on desire and trajectory only). NoBelief matches the full model on raw goal prediction (within 1 percentage point across both splits, averaged over 3 seeds) while NoDesire lags substantially, confirming that the belief channel imposes negligible predictive cost while enabling intervenable counterfactual reasoning; architecture details and full results are in Appendix C.

## 6.3 Desire Counterfactual Intervention

We evaluate desire interventions by replacing the inferred desire channel with category prototypes while holding belief and current-trajectory embeddings fixed. Figure 3 reports midpoint (50% path reveal) intervention results on the test and new-agent splits, with model-derived and handcrafted category-only prototypes shown

Table 2: Desire counterfactual significance at 50% path reveal. For each episode the mean off-diagonal $\Delta$ is computed as the average change in goal probability mass toward the transplanted category across all five non-matching prototype substitutions. Permutation $p$: one-sided random-sign test (H$_0$: mean $= 0$); sign test $p$: one-sided exact test (H$_0$: median $= 0$); CI: 95% bootstrap interval. Pairs sig.: off-diagonal (source, target) pairs with mean $\Delta > 0$ significant at $\alpha = 0.05$ after Holm–Bonferroni correction (30 pairs total).

| Split | Prototypes | $n$ | Mean $\Delta$ | 95% CI | Perm. $p$ | Sign test $p$ | Pairs sig. |
|---|---|---|---|---|---|---|---|
| Test | Model-derived | 480 | 0.151 | [0.145, 0.157] | $\leq 10^{-4}$ | $7.1 \times 10^{-136}$ | 29/30 |
| Test | Handcrafted | 480 | 0.344 | [0.335, 0.354] | $\leq 10^{-4}$ | $3.2 \times 10^{-145}$ | 30/30 |
| Test (new agent) | Model-derived | 480 | 0.135 | [0.129, 0.140] | $\leq 10^{-4}$ | $5.9 \times 10^{-138}$ | 27/30 |
| Test (new agent) | Handcrafted | 480 | 0.333 | [0.324, 0.341] | $\leq 10^{-4}$ | $3.2 \times 10^{-145}$ | 30/30 |

side-by-side for each split. Full path-fraction sweeps (25%, 50%, 75%, 100%) are provided in Appendix J; prototype vectors are in Appendix K.

Across prototype modes and splits, desire substitution induces coherent category-level shifts in the predicted goal distribution, evidenced by positive off-diagonal entries in the delta matrices: substituting a category prototype increases predicted probability mass on POIs in the target category. Concretely, if the desire channel of an episode with an ATTEND_CLASS desire is replaced with a GET_FOOD prototype, the goal head shifts probability toward food POIs, as expected from the causal BDI structure. This holds for both the model-derived and handcrafted prototype modes on both splits. Model-derived prototypes, which aggregate inferred desire vectors across many episodes and therefore distribute probability more broadly across all 24 POIs, produce smaller but still positive delta values compared to handcrafted prototypes.

Table 2 reports formal significance tests at 50% path reveal using the same battery as the belief counterfactuals: one-sided random-sign permutation tests, exact sign tests, and 95% bootstrap confidence intervals. For each episode we compute the mean off-diagonal $\Delta$ across all five non-matching prototype substitutions; we then test whether this pool of episode-level scalars has a positive mean. Results are strongly significant on all four combinations of split and prototype mode: permutation $p \leq 10^{-4}$ and sign-test $p < 10^{-130}$ in every case. At the pair level, 27–30 of the 30 off-diagonal (source, target) category pairs are individually significant at $\alpha = 0.05$ after Holm–Bonferroni correction, confirming that the positive off-diagonal pattern is not driven by one or two dominant substitutions.

### 6.4 Belief Counterfactual Intervention

Belief interventions are evaluated at 50% path reveal on episodes where the realized goal lies within the top-3 preference ranks. We set the original goal belief to closed and set same-category alternatives to open, while keeping desire and trajectory channels fixed. Figure 4 summarizes category-level effects for test and new-agent splits. We evaluate the same paired outcomes defined in Section 5.3: $\Delta_{\mathrm{alt}}$ (alternative gain) and $\Delta_{\mathrm{goal}}$ (goal drop), where positive $\Delta_{\mathrm{alt}}$ and negative $\Delta_{\mathrm{goal}}$ are the expected causal direction.

Table 3 reports episode-level inference. We use one-sided random-sign permutation tests for the mean effect (null: mean effect $= 0$), exact one-sided sign tests for directional consistency (null: 50/50 sign around zero), and 95% bootstrap confidence intervals for the mean. The primary causal test, $\Delta_{\mathrm{goal}}$, is strongly confirmed: sign tests are significant at $p < 10^{-6}$ on both splits, establishing that closing a goal POI's belief entry reliably reduces its predicted probability at the episode level. $\Delta_{\mathrm{alt}}$ is significant in mean but not sign-consistent, which is expected: the goal head redistributes dropped probability mass via learned desire and trajectory signals rather than applying a hard within-category reallocation, so some mass flows outside the target category. The intervention is also structurally asymmetric: 0–5 of 24 POIs are closed per belief configuration (3–5 for configurations with any false beliefs), so same-category alternatives are mostly already believed open in the original state; setting their belief entries to 1 therefore changes little, and the informative part of the intervention is concentrated on the goal POI being set to closed.

The 480 test episodes come from 15 agents with approximately 32 episodes each, but the episode-level tests above assume independence across observations from the same agent. A robustness sensitivity analysis that clusters by agent (one mean per agent and outcome, $n = 15$ agent-level means per split) is reported in

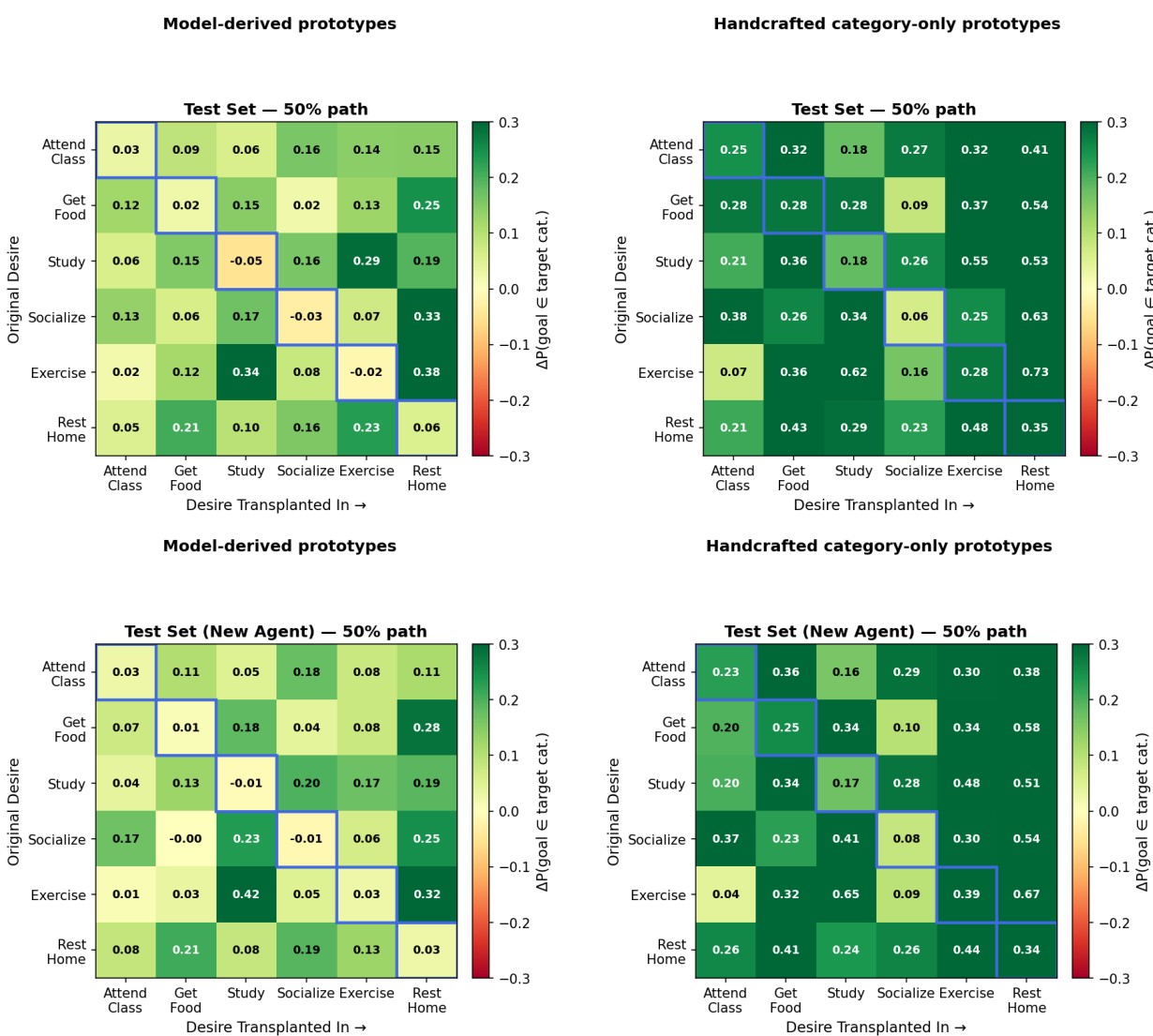

Figure 3: Desire counterfactual at 50% path reveal (2×2 panel). Top row: held-in test split; bottom row: held-out new-agent split. Left column: model-derived prototypes; right column: handcrafted category-only prototypes. Rows denote source desire category and columns denote transplanted category.

Table 3: Counterfactual belief significance summary at 50% path (rank ≤ 3). CI denotes 95% bootstrap confidence interval for the mean.

| Split | Outcome | $n$ | Mean | 95% CI | Permutation $p$ | Sign test $p$ |
|---|---|---|---|---|---|---|
| Test | $\Delta_{\mathrm{alt}}$ | 206 | 0.0061 | [0.0007, 0.0121] | 0.0194 | 0.1181 |
| Test | $\Delta_{\mathrm{goal}}$ | 206 | -0.0147 | [-0.0224, -0.0074] | $1.0 \times 10^{-4}$ | $2.93 \times 10^{-7}$ |
| Test (new agent) | $\Delta_{\mathrm{alt}}$ | 237 | 0.0055 | [0.0004, 0.0110] | 0.0238 | 0.0969 |
| Test (new agent) | $\Delta_{\mathrm{goal}}$ | 237 | -0.0240 | [-0.0299, -0.0183] | $5.0 \times 10^{-5}$ | $1.98 \times 10^{-18}$ |

Appendix F, Table 15. The directional pattern is preserved under clustering; however, held-in test effects weaken to borderline significance (goal-drop permutation $p = 0.052$), and the clearest retained effect is goal-drop on the held-out new-agent split (permutation $p = 0.0006$, sign test $p = 0.004$). The primary statistical evidence for the belief intervention therefore rests on the new-agent split, where the effect is robust to clustering. That being said, the 50% operating point understates the belief channel's influence at earlier reveals: at 25% path fraction, $\Delta_{\mathrm{goal}} = -0.020$ (test) and $-0.023$ (new agent), both larger in magnitude than

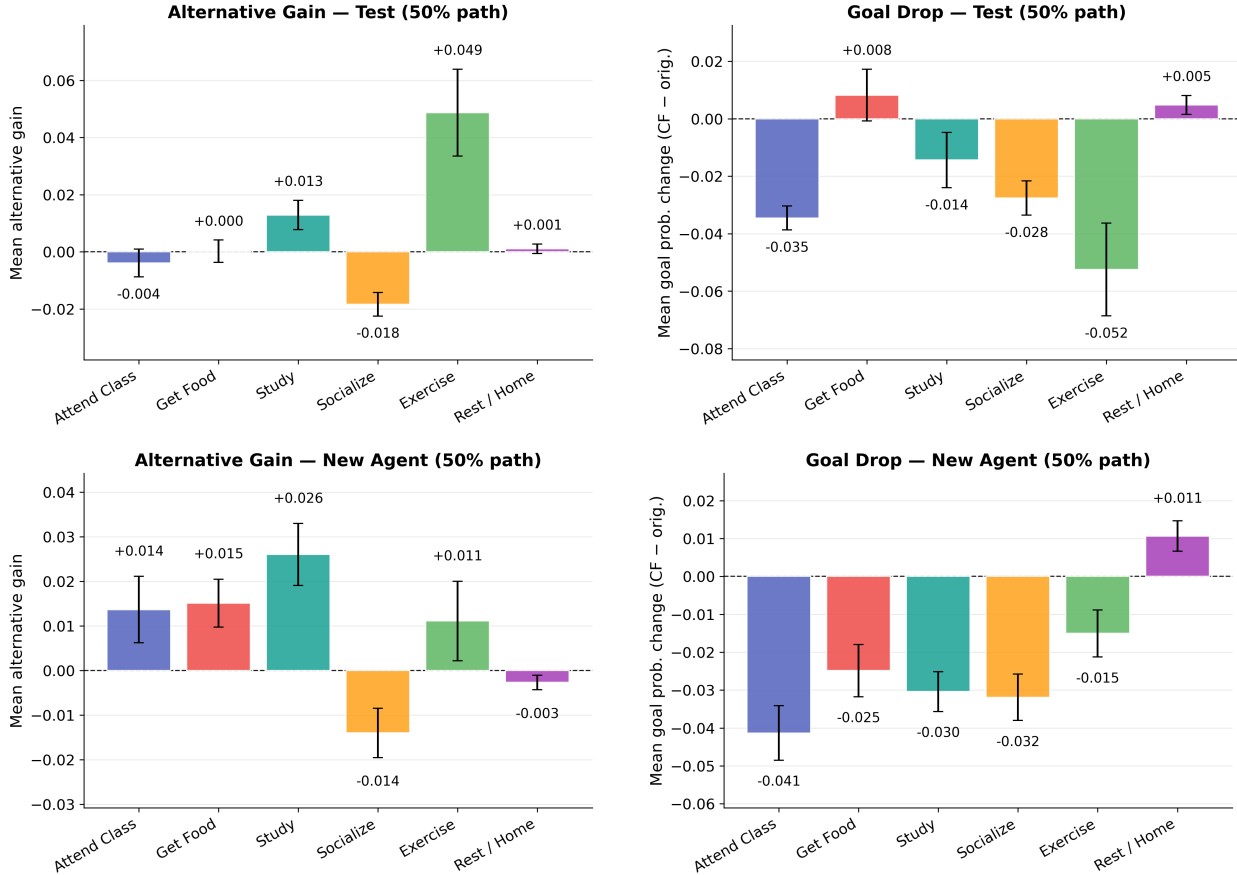

Figure 4: Belief counterfactual at 50% path reveal (rank $\leq 3$, 2×2 panel). Top row: held-in test episodes; bottom row: held-out new-agent episodes. Left column: mean alternative gain. Right column: mean goal drop. Error bars: $\pm 1$ SE.

the 50% values, consistent with intervention effects attenuating as trajectory evidence accumulates. Full path-fraction results are in Appendix F.7, Table 16.

Together, the desire and belief counterfactual results support the key claim that the intermediate channels are functionally intervenable: local edits propagate to goal prediction in the expected direction while preserving the intended modularity of the architecture. The desire channel produces large and consistent category-level shifts. The belief channel produces smaller but directionally reliable goal-probability drops, especially on held-out new agents and at earlier path reveals; its limitations and supporting diagnostics are analyzed in the appendix. Section 6.5 moves from this local controllability to the harder question of whether these edits also predict behavior under a regenerated future path.

## 6.5 Trajectory-Level Counterfactual Consistency

The previous interventions do not show that edits to the desire and belief channels predict an agent's next actions. We test this external validity with the model-planned diagnostic (Section 5.3): at 50% prefix, we edit the model's inferred channel and then read its counterfactual goal $g_{\mathrm{cf}}$; we independently ask the simulator, placed in the same edited mental state, which goal $g_{\mathrm{sim}}$ its agent would pursue. Table 4 reports agreement between the two; full probability-delta tables and cluster significance tests are in Appendix F.8.

Two findings hold across both splits. First, the model's counterfactual goal agrees with the agent's actual counterfactual goal well above chance: top-5 agreement is 33–47% (shuffled-goal $p_{\mathrm{shuffle}} = 0.0016$ in every cell), strongest for desire edits in the dense regime (43.3% held-in, 47.2% new-agent). Exact top-1 agreement

Table 4: Model-planned versus simulator-resampled counterfactual agreement at 50% path reveal. Top-1 is the percentage of cases where the model-selected counterfactual goal equals the simulator-resampled goal; Top-5 is the percentage where the simulator goal appears in the model's edited top-5. Values are means with 95% bootstrap confidence intervals clustered by seed-agent. Top-5 agreement exceeds a within-group shuffled-goal baseline in every condition (Holm-adjusted permutation $p \leq 0.0016$).

| Regime | Split | Type | $n$ | Top-1 (%) | Top-5 (%) |
|--------|-------|------|-----|-----------|-----------|
| dense | test | belief | 2880 | 6.4 [4.7, 8.1] | 34.7 [29.3, 40.6] |
| dense | test | desire | 2880 | 18.1 [15.4, 20.9] | 43.3 [40.1, 46.4] |
| dense | test_new_agent | belief | 2880 | 7.7 [6.2, 9.2] | 36.8 [32.5, 41.0] |
| dense | test_new_agent | desire | 2880 | 21.5 [18.6, 24.4] | 47.2 [44.2, 50.1] |
| sparse | test | belief | 2400 | 8.9 [7.4, 10.4] | 37.2 [33.7, 41.2] |
| sparse | test | desire | 2400 | 9.0 [7.5, 10.6] | 33.4 [30.5, 36.4] |
| sparse | test_new_agent | belief | 2400 | 8.2 [6.5, 10.0] | 35.1 [31.7, 38.5] |
| sparse | test_new_agent | desire | 2400 | 9.5 [8.0, 11.0] | 36.0 [32.9, 39.0] |

is lower (6–22%), as expected when several counterfactual futures remain plausible from a half-observed prefix. Second, the intervention moves probability in the causally correct direction: the model-selected goal reliably gains mass over the original-goal baseline ($\Delta_{\mathrm{sel}} > 0$, cluster $p_{\mathrm{adj}} = 0.003$ in every condition), and the original goal loses mass ($\Delta_{\mathrm{orig}} < 0$) everywhere except sparse belief, where closing a single believed-open goal does not reliably suppress it ($p_{\mathrm{adj}} = 1.0$).

This more-global analysis mirrors the local results: desire interventions transfer to predicted behavior most strongly, while belief interventions transfer in the dense regime but are weaker when false beliefs are sparse. Exact trajectory-level agreement remains limited, so we read this as controlled counterfactual *consistency*, not as establishing full behavioral counterfactual prediction.

## 7 Discussion

Our analyses reveal significant positive results for the BDIBOTTLENECK framework, though with clear limitations. The simulation-based design is a methodological necessity: real-world behavioral datasets contain no ground-truth belief or desire labels, so attribution claims cannot be validated on real data without first establishing that the model recovers the correct latent variables in a controlled setting. External validity to real mobility behavior remains an important open question, though one the current results substantially inform. In addition, several inferential effects weaken under agent-clustered sensitivity analysis, indicating that repeated samples per agent can amplify episode-level significance. Finally, the belief configurations in the primary simulation close 0–5 POIs out of 24 per episode (3–5 in configurations with any false beliefs), creating a sparse and imbalanced signal for both belief head training and counterfactual evaluation. The dense false-belief diagnostic in Appendix F shows that closed-entry F1 improves when the simulator supplies more minority-class belief evidence, but it also confirms that belief remains the harder channel. Belief effects should therefore be read as real but weaker than desire effects, and especially weaker than the trajectory signal at later path reveals.

The counterfactual experiments also have different scopes. The main desire and belief interventions are local channel interventions: they edit $\hat{\pi}$ or $\hat{b}$ while keeping the observed prefix fixed, then re-evaluate the frozen goal head. These experiments establish intervenability of the learned representation, not full behavioral counterfactual identification. The trajectory-level diagnostics are a stricter check: the model-planned analysis (Section 6.5) compares the edited model's predicted goal to the goal the simulator's agent would actually pursue, and simulator-resampled rollouts in the appendix add a full-trajectory behavioral target. These diagnostics strengthen the positive evidence, especially for desire and for dense belief settings, but they also show that full counterfactual behavior prediction is harder than local controllability.

The most important next step is transfer to richer settings: simulators with dynamic world states and stronger heterogeneity, and then real-world behavioral traces with explicit proxy measures for latent states. Full supervision is not available in real-world settings, so future work will likely continue to require simulations that define and test belief-desire-intention proxy variables from contextual information and trajectories to identify those that are actually predictive of the agent's true internal state. Progress on this step will require careful

protocol design for proxy-label construction, robustness checks for counterfactual claims, and calibration diagnostics under distribution shift.

Consider some concrete examples of potential proxy definitions. Desire proxies are the easier case, since a visited destination is a revealed preference: clustering historical destinations by location and activity type yields candidate goals with little labeling effort. Belief proxies are harder since beliefs are never directly observed. However, behavior still leaks information about beliefs. For example, route choices that change only after a service disruption is publicly announced can indicate whether an agent currently believes a location is available. Such signals are noisy on a per-individual basis, but potentially usable in aggregate to supervise belief head training. Whatever the proxy, validation is essential, and our counterfactual protocol offers a template: instead of checking latent states directly, test whether intervening on them shifts predicted behavior as domain knowledge predicts, using natural experiments such as disruptions with known announcement times in place of ground-truth labels.

Scaling to more complex environments will be uneven across components. Larger or continuous spaces are inexpensive. The trajectory encoders operate on raw normalized coordinates rather than graph node indices or adjacency, so the same stack can ingest longer or denser sequences without modification. The binding constraint is the output side: the belief and desire heads are fixed-cardinality projections onto the 24 known POIs, so an open-ended location set would require replacing them with a set-based, pointer-style formulation that embeds and scores each candidate. Dynamic environments requiring sequential belief updates would require running the belief head as an online filter that revises its estimate as each new observation arrives, rather than the current design in which beliefs are inferred once per episode from static context.

We introduced SMToM and instantiated it as BDIBOTTLENECK, demonstrating that explicit BDI-structured supervision can support goal inference while exposing belief and desire channels that can be intervened on directly. The strongest supported conclusion is calibrated: in a controlled navigation setting, auxiliary supervision accounts for part of the raw goal-prediction gain, so the bottleneck's distinctive contribution is not black-box dominance but explicit mental-state structure that makes counterfactual analysis native to the model. Desire interventions producing robust goal shifts and belief interventions producing smaller but measurable effects whose strength depends on the density of false-belief evidence.

**Broader Impact Statement**

Machine learning models that infer mental states such as emotions, goals, and beliefs can easily invade privacy and cause direct harm to the people they model. For example, mobility traces are already collected at scale, and a model that reliably attributes latent beliefs or desires to a person's movements could enable a state or corporate actor to predict dissent, ill health, or other private features without the individual's knowledge or consent. In fact, the EU AI Act explicitly bans AI systems that infer emotional or mental state information in workplace and educational contexts. Any deployment must thus be evaluated on not only accuracy, but also whether (i) predictions could harm an individual, whether accurate or not; (ii) individuals can seek recourse; and (iii) there is general awareness that inference of mental states could occur. We view these models as most appropriate in settings that plausibly satisfy these criteria while maintaining limited scope. For example, additional context about inferred mental states could improve a service for the population being modeled, such as urban planning (e.g., asking how ridership would change if a population believed the subway was safe) or robotics (inferring a person's goal to assist rather than to track or profile them). We flag this distinction as a necessary consideration for any future deployment of SMToM-style models on real-world trajectory data, rather than as a resolution to it.

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

## Appendix A   Dataset Split Sizes

Table 5: Dataset split sizes used for training and evaluation.

| Split | Configs | Agents | Total episodes |
|---|---|---|---|
| `train` | $B_0-B_7$ (8) | 15 | 2040 |
| `val` | $B_0-B_7$ (8) | 15 | 480 |
| `test` | $B_0-B_7$ (8) | 15 | 480 |
| `test_new_agent` | $B_0-B_7$ (8) | 15 | 480 |
| `test_bh` | BH0, BH1 (2) | 15 | 480 |
| `test_new_agent_bh` | BH0, BH1 (2) | 15 | 480 |

## Appendix B   Training Details and Hyperparameters

Table 6 reports the exact training configuration used for the original three neural models. Table 7 reports the matched auxiliary-supervised context baseline added to isolate auxiliary supervision from the BDI bottleneck. Table 8 reports the settings specific to the two BDIBOTTLENECK ablations. Unless otherwise noted, values are taken directly from script defaults; the global override used in the paper-facing goal-inference runs is training duration (300 epochs for every model).

Table 6: Per-model training specification used in reported experiments.

| Setting | GOALPREDICTOR | CONTEXT GOALPREDICTOR | BDIBOTTLENECK |
|---|---|---|---|
| Training script | `train_goal_predictor.py` | `train_context_goal _predictor.py` | `train_bdi_bottleneck.py` |
| Epochs | 300 | 300 | 300 |
| Batch size | 64 | 32 | 32 |
| Optimizer | AdamW | AdamW | AdamW |
| Base learning rate | $3 \times 10^{-4}$ | $3 \times 10^{-4}$ | $3 \times 10^{-4}$ |
| Weight decay | $10^{-4}$ | $10^{-4}$ | $10^{-4}$ |
| LR schedule | Cosine annealing; $\eta_{\min} = \eta_0/10$ | Warmup 0, then cosine; $\eta_{\min} = \eta_0/10$ | Warmup 0, then cosine; $\eta_{\min} = \eta_0/10$ |
| Gradient clipping | Global norm 1.0 | Global norm 1.0 | Global norm 1.0 |
| Dropout | 0.1 | 0.1 | 0.1 |
| Transformer config | $d_{\text{model}} = 64$, layers=3, heads=4, dim_ff=256 | $d_{\text{model}} = 64$, layers=3, heads=4, dim_ff=256 | $d_{\text{model}} = 64$, layers=3, heads=4, dim_ff=256 |
| Context sizes | Not used | $K = 10$, $K_{\text{recent}} = 5$ | $K = 10$, $K_{\text{recent}} = 5$ |
| Trajectory augmentation | Random prefix truncation | Random prefix truncation | Random prefix truncation |
| Goal loss | Cross-entropy | Cross-entropy | $\lambda_g \cdot$ cross-entropy ($\lambda_g = 1.0$) |
| Desire loss | Not used | Not used | $\lambda_d \cdot$ KL divergence ($\lambda_d = 1.0$) |
| Belief loss | Not used | Not used | $\lambda_b \cdot$ weighted BCE ($\lambda_b = 5.0$, false-belief weight = 5.0) |
| Checkpoint criterion | Best val goal top-1 | Best val goal top-1 | Best val goal top-1 |
| Eval checkpoints | Best checkpoint only | Best checkpoint only | Best checkpoint only |

The fair auxiliary-supervised baseline uses the same context streams and the same auxiliary desire/belief losses as BDIBOTTLENECK, but the goal head reads the raw encoder embeddings. This is the relevant control for the concern that BDIBottleneck's raw goal accuracy could improve simply because it receives extra multi-task supervision. If AuxContext matches or exceeds BDIBottleneck on raw accuracy, the correct interpretation is not that the bottleneck is unnecessary; it is that auxiliary supervision accounts for part of the predictive gain, while the bottleneck's distinctive contribution is a directly editable belief/desire interface for counterfactual analysis.

Table 7: Training specification for the matched auxiliary-supervised context baseline. Settings are intentionally aligned with BDIBOTTLENECK; the only architectural difference is the goal-head input.

| Setting | AUXCONTEXT |
|---|---|
| Training script | `train_context_goal_predictor_aux.py` |
| Epochs, seeds, context | 300 epochs; seeds 0, 5, 10; $K = 10$, $K_{\text{recent}} = 5$ |
| Architecture | Same three transformer encoders as CONTEXTGOALPREDICTOR and BDIBOT-TLENECK; $d_{\text{model}} = 64$, layers=3, heads=4, dim_ff=256, dropout=0.1 |
| Goal head | $\text{MLP}_{\text{aux}}([z_d\|z_r\|z_\tau])$; no belief/desire bottleneck in the goal path |
| Auxiliary heads | Desire head on $z_d$ and belief head on $[z_d\|z_r]$, matching BDIBOTTLENECK's targets |
| Losses | Goal cross-entropy, desire KL, and weighted belief BCE with $\lambda_g = 1.0$, $\lambda_d = 1.0$, $\lambda_b = 5.0$ and false-belief weight 5.0 |
| Optimizer and selection | AdamW, learning rate $3 \times 10^{-4}$, weight decay $10^{-4}$, cosine schedule, best checkpoint selected by validation goal top-1 |

Figure 5 shows validation diagnostic curves for BDIBOTTLENECK across all 300 training epochs. Belief overall accuracy (left panel) rises steadily and plateaus around 76%, though this figure is dominated by the majority open-class entries. False-belief F1 on the closed-entry subset (center panel) peaks at approximately 40% in early training and declines to 33% at the best checkpoint, reflecting the multi-task objective's increasing emphasis on goal accuracy as training proceeds. Desire KL divergence (right panel) drops rapidly from 0.69 to below 0.30 within the first 50 epochs and continues declining to approximately 0.22, indicating convergent and stable desire attribution throughout training.

Figure not found. Expected at `outputs/bdi_bottleneck/training_curves.png`.

Figure 5: BDIBOTTLENECK validation diagnostics across 300 training epochs. Left: overall belief accuracy (all 24 entries); inflated by the majority open class. Center: false-belief F1 on believed-closed entries only; peaks at 40.4% early and converges to 33.3%; an all-open predictor achieves 0% F1, so the converged value reflects genuine false-belief discrimination. Right: desire KL divergence against ground-truth preference distribution; drops from 0.69 to 0.22, reflecting rapid and stable desire learning. Best checkpoint is selected by validation goal top-1.

Table 8 reports the settings specific to the two ablations. All hyperparameters not listed (optimizer, base learning rate, weight decay, LR schedule, gradient clipping, dropout, transformer architecture, batch size) are identical to BDIBOTTLENECK as given in Table 6.

Table 8: Training settings specific to the two ablation variants. Unlisted hyperparameters match BDIBOTTLENECK exactly (Table 6).

| Setting | BDIBOTTLENECK-NODESIRE | BDIBOTTLENECK-NOBELIEF |
|---|---|---|
| Training script | `train_bdi_bottleneck_ablation.py -ablation no_desire` | `train_bdi_bottleneck_ablation.py -ablation no_belief` |
| Epochs | 300 | 300 |
| Context sizes | $K_{\text{recent}} = 5$; desire context $K$ not used | $K = 10$; recent context $K_{\text{recent}}$ not used |
| Goal loss | $\lambda_g \cdot$ cross-entropy ($\lambda_g = 1.0$) | $\lambda_g \cdot$ cross-entropy ($\lambda_g = 1.0$) |
| Desire loss | Not used | $\lambda_d \cdot$ KL divergence ($\lambda_d = 1.0$) |
| Belief loss | $\lambda_b \cdot$ weighted BCE ($\lambda_b = 5.0$, false-belief weight = 5.0) | Not used |
| Checkpoint criterion | Best val goal top-1 | Best val goal top-1 |

## Appendix C    BDIBottleneck Ablation Study

**Architecture**

BDIBOTTLENECK-NODESIRE removes the desire encoder and desire head entirely. The belief head is simplified to $\text{Linear}(d_{\text{model}}, M)$, reading only the recent-context embedding $z_r$ rather than the concatenated $[z_d \| z_r]$ used in the full model. The goal head input is correspondingly reduced to $[\hat{b} \| z_\tau] \in \mathbb{R}^{M+d_{\text{model}}}$. This model receives no agent preference information beyond what the trajectory prefix itself implies.

BDIBOTTLENECK-NOBELIEF removes the recent encoder and belief head entirely. The goal head input is $[\hat{\pi} \| z_\tau] \in \mathbb{R}^{M+d_{\text{model}}}$. This model has no representation of which POIs the agent believes to be currently available, and therefore cannot support belief counterfactual interventions.

Both ablations retain the BDI bottleneck structure: the goal head reads predicted probability outputs ($\hat{\pi}$ or $\hat{b}$) rather than raw transformer embeddings, so any performance differences reflect the marginal value of the removed channel rather than a confound from changing the goal head's input type.

**Goal Inference Results**

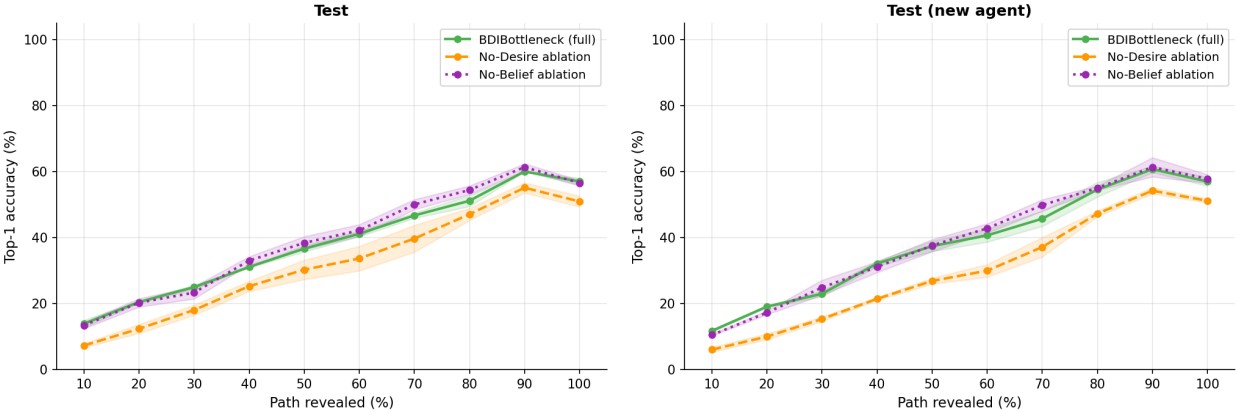

Figure 6: Top-1 goal accuracy vs. path fraction for BDIBOTTLENECK (full), BDIBOTTLENECK-NODESIRE, and BDIBOTTLENECK-NOBELIEF. Left: held-in test split. Right: held-out new-agent split. NOBELIEF closely tracks the full model at all fractions; NODESIRE lags substantially, demonstrating that desire context is the primary driver of goal disambiguation. Shaded bands: $\pm 1$ SD across 3 training seeds.

BDIBOTTLENECK-NOBELIEF closely tracks the full BDIBOTTLENECK on top-1 goal accuracy across all path fractions and both evaluation splits, and is marginally higher at several fractions. BDIBOTTLENECK-NODESIRE lags substantially below both at every fraction, with the gap widening at shorter path reveals where trajectory evidence is weakest and the preference prior is most important for disambiguation.

**Interpretation**

The pattern is consistent with the BDI causal structure of the environment. Desire encodes agent category preferences over all 24 POIs and is the dominant source of prior information for goal prediction; without it, the model has no basis for preference-based disambiguation beyond the observed trajectory prefix. Belief, by contrast, only modulates which POIs in the preferred category are currently believed available. Since only 0–5 of 24 POIs are believed closed per episode (3–5 in configurations with any false beliefs), the belief state affects a small fraction of the goal distribution in expectation, so its marginal contribution to raw prediction accuracy is modest.

The near-parity of NOBELIEF and the full model on prediction is not a weakness of the belief channel; it reflects the task structure and the intended role of that component. The belief channel's purpose is to provide an interventionable representation that supports counterfactual analysis, not to maximize marginal predictive accuracy. This distinction is important: BDIBOTTLENECK-NOBELIEF can match the full model on goal prediction but cannot answer the question *what would this agent do if they believed a specific location was closed?* There is no belief slot to intervene on. The counterfactual evidence in Section 6 demonstrates that the full model's belief slot is both present and causally active (significant $\Delta_{\text{goal}}$ with $p < 10^{-6}$), which is precisely what the ablation cannot achieve.

The small accuracy cost of retaining the belief channel therefore buys a discrete and non-trivial capability: the ability to pose and answer belief counterfactuals over inferred agent mental states. Viewed through the lens of the interpretable-machine-learning literature, this is the standard interpretability–accuracy tradeoff instantiated at the level of mental-state components. Models constrained to produce human-interpretable intermediate representations often match or nearly match their unconstrained counterparts on prediction while providing structure that cannot be recovered post hoc from black-box outputs. BDIBOTTLENECK instantiates this principle at the level of BDI components: explicit belief and desire slots are maintained throughout the forward pass so that interventions on those slots are a native operation, not a retrofit.

## Appendix D    Additional Problem and Loss Details

This section collects implementation-level assumptions and secondary formulas moved from the main text for space.

**Belief-feasible goal set.**   For sampled desire category $d$, let $\mathcal{V}(d) \subset \mathcal{V}_{\text{POI}}$ be the POIs in category $d$. The simulator defines the belief-feasible set

$$\mathcal{A}(d, b) = \{v \in \mathcal{V}(d) : b_v = 1\}. \tag{7}$$

If $\mathcal{A}(d, b) = \emptyset$, the simulator resamples $d$; otherwise it samples a goal from the within-category preference distribution restricted to $\mathcal{A}(d, b)$ and then samples a route via a Boltzmann distribution over $k$ shortest paths.

**Auxiliary model details.**   In reported runs, transformer encoders use $d_{\text{model}} = 64$, 3 layers, 4 heads, and feedforward width 256. The goal MLP is two-layer with hidden width 256 and ReLU.

**Per-term losses.**   Using

$$\hat{p}_g = \text{softmax}(\ell_g), \qquad \hat{p}_d = \text{softmax}(\ell_d), \qquad \hat{p}_b = \sigma(\ell_b), \tag{8}$$

the per-sample terms are

$$\mathcal{L}_{\text{goal}} = -\log \hat{p}_g[g], \tag{9}$$

$$\mathcal{L}_{\text{desire}} = \text{KL}(\pi \,\|\, \hat{p}_d) = \sum_{j=1}^{M} \pi_j \log \frac{\pi_j}{\hat{p}_{d,j}}, \tag{10}$$

and weighted belief BCE

$$\mathcal{L}_{\text{belief}} = -\frac{1}{M} \sum_{j=1}^{M} w_j \big[ b_j \log \hat{p}_{b,j} + (1 - b_j) \log(1 - \hat{p}_{b,j}) \big], \tag{11}$$

with $w_j = \omega$ for $b_j = 0$ and $w_j = 1$ for $b_j = 1$.

Defaults in the training scripts are $\lambda_g = 1$, $\lambda_d = 1$, $\lambda_b = 5$, and $\omega = 5$.

## Appendix E   Anonymized Simulation Configuration Details

This section reports the exact belief-configuration and character-parameter settings used by the simulator, with all POI names replaced by anonymized identifiers. POIs are indexed as `POI-01` through `POI-24` by category order and within-category order.

Table 9: Anonymized POI indexing used in this appendix.

| Category | POI IDs (within-category order) |
|---|---|
| attend_class | POI-01, POI-02, POI-03, POI-04 |
| get_food | POI-05, POI-06, POI-07, POI-08 |
| study | POI-09, POI-10, POI-11, POI-12 |
| socialize | POI-13, POI-14, POI-15, POI-16 |
| exercise | POI-17, POI-18, POI-19, POI-20 |
| rest_home | POI-21, POI-22, POI-23, POI-24 |

Belief configurations are generated from `config/belief_configs.json`. Each config specifies a set of falsely-believed-closed POIs; all others are believed open. Episodes per config are 25.

Table 10: Belief configurations (anonymized POI IDs).

| Config | Split | False-belief POIs (believed closed) |
|---|---|---|
| B0 | train | none |
| B1 | train | POI-09, POI-10, POI-11 |
| B2 | train | POI-05, POI-06, POI-07 |
| B3 | train | POI-17, POI-18, POI-19 |
| B4 | train | POI-13, POI-14, POI-15 |
| B5 | train | POI-09, POI-10, POI-05, POI-06 |
| B6 | train | POI-17, POI-18, POI-19, POI-13, POI-14 |
| B7 | train | POI-01, POI-03, POI-02, POI-09, POI-10 |
| BH0 | held-out | POI-09, POI-11, POI-17, POI-18, POI-19 |
| BH1 | held-out | POI-05, POI-06, POI-08, POI-15, POI-16 |

Character parameters are taken from `config/agent_characters.json`. Table 11 reports category-level Dirichlet parameters. Table 12 reports within-category Dirichlet parameters (4-vector aligned to the POI order in Table 9).

Table 11: Category-level Dirichlet parameters by character (category order: attend_class, get_food, study, socialize, exercise, rest_home).

| Character | class | food | study | social | exercise | home |
|---|---|---|---|---|---|---|
| studious_stem | 3.0 | 1.0 | 5.0 | 0.3 | 0.4 | 0.8 |
| social_athlete | 0.5 | 2.0 | 0.4 | 4.5 | 4.0 | 0.6 |
| foodie_socializer | 0.8 | 5.0 | 0.8 | 3.5 | 0.5 | 0.4 |
| homebody_studier | 1.0 | 0.6 | 2.5 | 0.2 | 0.2 | 6.0 |
| well_rounded | 2.0 | 2.0 | 2.0 | 2.0 | 1.5 | 1.5 |

Table 12: Within-category Dirichlet parameters by character. Each tuple is a 4-vector aligned to the category's POI order in Table 9.

| Character | Node alphas by category |
|---|---|
| studious_stem | class (1.5, 1.0, 3.0, 0.5); food (0.5, 2.5, 3.0, 0.5); study (1.5, 3.5, 1.0, 3.0); social (1.5, 1.0, 0.5, 1.5); exercise (1.0, 2.0, 0.5, 1.5); home (1.0, 3.0, 1.0, 1.5) |
| social_athlete | class (2.0, 1.0, 1.0, 0.5); food (3.5, 0.5, 0.5, 3.0); study (2.0, 0.5, 2.5, 0.5); social (2.5, 1.0, 3.5, 1.5); exercise (4.0, 1.0, 1.5, 3.5); home (3.0, 1.0, 0.5, 2.0) |
| foodie_socializer | class (1.0, 1.5, 1.0, 1.5); food (3.0, 2.0, 1.0, 4.0); study (2.5, 0.5, 2.0, 0.5); social (1.5, 4.0, 1.0, 4.5); exercise (1.0, 1.0, 0.5, 2.5); home (2.0, 1.0, 2.0, 1.0) |
| homebody_studier | class (1.0, 1.5, 1.0, 0.5); food (0.5, 4.0, 1.0, 0.5); study (3.5, 1.0, 0.5, 2.0); social (1.0, 2.0, 0.5, 1.0); exercise (0.5, 1.5, 0.5, 2.0); home (1.0, 1.0, 4.0, 2.5) |
| well_rounded | class (2.0, 2.0, 2.0, 1.5); food (2.0, 2.0, 2.0, 2.0); study (3.0, 1.5, 2.0, 1.5); social (2.0, 2.0, 1.5, 2.0); exercise (2.5, 2.0, 1.5, 2.0); home (2.0, 2.0, 2.0, 1.5) |

## Appendix F    Additional Experiments and Results

### F.1    Sparse and Dense False-Belief Regimes

The main simulator uses a deliberately sparse false-belief regime: each episode has a 24-entry belief vector over POIs, and the training configurations close either no POIs or a small subset of 3–5 POIs. This is the setting used for the main results because it reflects the intended navigation problem: most places are normally available, and false beliefs are exceptional. The consequence is a severe class imbalance for the belief head. Across sparse test episodes, roughly 13–16% of belief entries are closed and the remainder are open, so an uninformative all-open predictor can obtain high overall accuracy while having zero false-belief F1.

To diagnose whether low false-belief F1 reflects a fundamental inability of the architecture or the scarcity of closed labels, we also constructed a dense false-belief regime. Dense training keeps the original sparse configurations $B_0$–$B_7$ and adds medium configurations BM1–BM4 with 6–8 closed POIs and dense configurations BD1–BD4 with 11–12 closed POIs. Dense held-out evaluation uses BHD0 and BHD1 with 12 closed POIs. Dense beliefs are therefore a diagnostic stress test with more supervision on the minority closed class; they are not a replacement for the sparse benchmark.

Table 13: Sparse versus dense false-belief regimes. Closed-entry F1 and PR-AUC evaluate the minority believed-closed class, not the majority open class.

| Regime | Configurations | Closed-entry rate | Closed F1 | Closed PR-AUC |
|---|---|---|---|---|
| Sparse test | $B_0$–$B_7$ | 0.135 | 0.390 | 0.300 |
| Sparse new agent | $B_0$–$B_7$ | similar sparse rate | 0.396 | 0.309 |
| Dense test | $B_0$–$B_7$, BM1–BM4, BD1–BD4 | 0.298 overall | 0.524 | 0.462 |
| Dense new agent | same dense family | 0.298 overall | 0.521 | 0.453 |

Table 13 shows the central diagnostic result. Under the sparse benchmark, false-belief F1 is around 0.39 despite high overall belief accuracy, confirming that the original belief head is learning a real but imperfect minority-class signal. Under denser false-belief supervision, closed-entry F1 rises to approximately 0.52 and PR-AUC rises to approximately 0.46. This supports the interpretation that the weak sparse false-belief F1 is largely a data-regime limitation caused by rare closed labels and not simply a broken belief architecture. At the same time, the dense regime does not eliminate the limitation: belief remains harder and less stable than desire, so the paper should present belief evidence as weaker but nonzero.

### F.2    Same-Budget Robustness and Belief-Loss Variants

We ran a same-budget robustness sweep for BDIBOTTLENECK and AUXCONTEXT. In the sparse regime, both models were trained across five seeds (0, 5, 10, 15, 20) and four pre-registered configurations: the default weighted BCE used in the paper, a lower-learning-rate weighted BCE, balanced BCE, and balanced focal loss. The balanced losses compute a batch-level closed-entry weight from the open-to-closed ratio, clamped between 1 and 12; balanced focal additionally multiplies the BCE term by $(1 - p_t)^2$ to emphasize hard belief entries. In the dense confirmation, both models were trained across seeds 0, 5, and 10 for the default configuration and the preselected belief-optimized configuration, balanced focal. The selection rule was fixed before dense training: choose the non-default sparse configuration with the highest validation closed-class F1 among configurations whose validation goal top-1 is within 0.02 of default; balanced focal satisfied this rule.

Table 14: Same-budget robustness summary. Values are mean ± standard deviation across seeds on the test split. The sweep is diagnostic rather than a tuned winner selection.

| Regime | Config | Model | Top-1 | Closed F1 |
|--------|--------|-------|-------|-----------|
| Sparse | default weighted | AuxContext | $0.253 \pm 0.013$ | $0.399 \pm 0.006$ |
| Sparse | default weighted | BDIBottleneck | $0.196 \pm 0.036$ | $0.392 \pm 0.006$ |
| Sparse | balanced focal | AuxContext | $0.248 \pm 0.008$ | $0.389 \pm 0.004$ |
| Sparse | balanced focal | BDIBottleneck | $0.195 \pm 0.052$ | $0.387 \pm 0.008$ |
| Dense | default weighted | AuxContext | $0.302 \pm 0.009$ | $0.524 \pm 0.002$ |
| Dense | default weighted | BDIBottleneck | $0.305 \pm 0.064$ | $0.523 \pm 0.003$ |
| Dense | balanced focal | AuxContext | $0.312 \pm 0.008$ | $0.529 \pm 0.002$ |
| Dense | balanced focal | BDIBottleneck | $0.307 \pm 0.036$ | $0.525 \pm 0.005$ |

The robustness sweep sharpens the interpretation of Table 1. In the sparse regime, AuxContext often has higher raw top-1 accuracy than BDIBottleneck under the same auxiliary losses, indicating that some predictive information is easier to exploit from the raw context embeddings than through the explicit bottleneck. In the dense regime, the two models are much closer on raw accuracy and closed-entry F1. The supported claim is therefore not that BDIBottleneck is always the strongest black-box predictor. The supported claim is that BDIBottleneck remains competitive while exposing belief and desire variables as editable channels, enabling the counterfactual tests that AuxContext cannot perform directly.

### F.3 Belief Channel Use and Sensitivity Diagnostics

The belief channel was probed in four ways. First, direct attribution metrics measured closed-entry F1 and PR-AUC, reported in Table 13. Second, channel-use masking compared the normal model to variants where belief was set all-open, shuffled, oracle-injected, or randomized. Third, dose-response diagnostics gradually interpolated the belief intervention strength from the original channel to the edited channel. Fourth, gradient and goal-head weight analyses measured whether the goal loss is sensitive to the belief entries.

The sparse model shows a small but measurable belief contribution. At 50% reveal on the test split, setting all beliefs open changes the goal distribution with mean KL around 0.024, while shuffling belief produces much smaller KL around 0.002. In the dense regime, the corresponding all-open KL rises to about 0.054 and shuffle KL to about 0.012, indicating stronger dependence on belief when the data provide denser closed-entry evidence. Goal-head input norms are also nonzero for belief: the sparse model has approximate goal-head norms 9.01 for desire, 5.88 for belief, and 9.46 for trajectory; the dense model has 8.67 for desire, 5.93 for belief, and 9.57 for trajectory. These diagnostics support the claim that the belief channel is used, but less strongly than desire and trajectory.

The dose-response evidence is similarly calibrated. In sparse data, belief-dose slopes are small and sometimes noisy; in dense data, the effect becomes more monotone and more positive. This is consistent with the false-belief class imbalance: when only a few of 24 entries are closed, most belief interventions alter a small part of the goal distribution, and the learned goal head can often rely on desire and trajectory instead. Dense false-belief supervision increases the signal available to the belief head and makes its downstream effect easier to detect.

### F.4 Distractor Robustness

This evaluation tests whether the model can maintain correct goal ranking when the agent's trajectory passes spatially near a low-preference POI en route to a high-preference destination. We refer to such a POI as a *distractor*: it is geometrically proximate to the path but causally irrelevant to the agent's goal.

Episodes are filtered to a distractor subset using two criteria: (1) the agent's true goal is among its top-$K$ preferred POIs by desire vector and (2) at least one bottom-$K$ POI lies within a threshold distance of some path node, excluding the final segment. Default parameters are $K = 5$ and threshold = 50 metres. For each qualifying episode, we identify the *distractor moment*: the path step at which the agent is closest to the

distractor POI. This is the hardest inference point because the trajectory prefix, truncated to that step, is spatially most consistent with the distractor as the destination.

All three models are evaluated at the distractor moment on the same truncated trajectory prefix. The headline metric is the percentage of distractor episodes in which each model correctly ranks the true goal above the distractor POI in its predicted goal distribution, reported per belief configuration and per split.

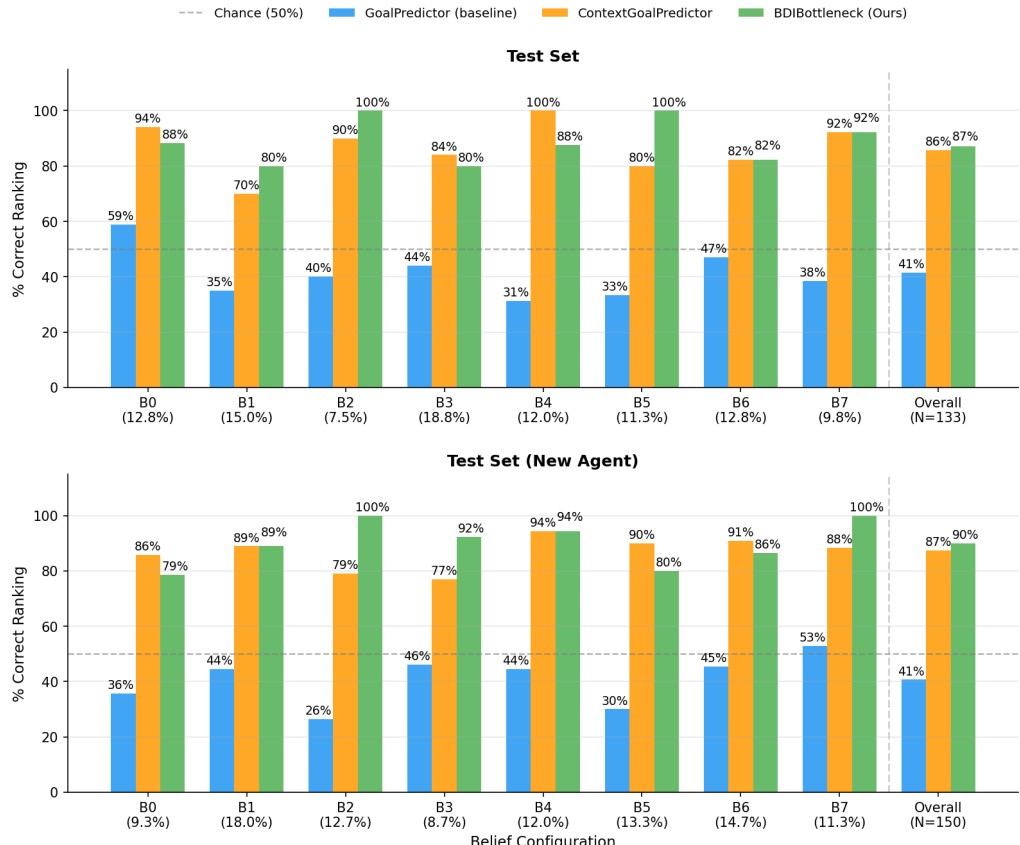

Figure 7: Percentage of distractor episodes where the model ranks the true goal above the distractor POI, by belief configuration. B0 is excluded (no false beliefs). The dashed line marks chance (50%). Upper panel: held-in test agents. Lower panel: new agents.

Aggregating over all distractor episodes and belief configurations, BDIBOTTLENECK ranks the true goal above the distractor more often than GOALPREDICTOR and CONTEXTGOALPREDICTOR on both splits. Per-configuration results are mixed: no model dominates across every configuration, reflecting the small episode count per configuration in the distractor subset. The aggregate advantage is consistent with the model having learned agent-level preference representations that override spatial proximity; a model with no access to preference context would have no basis for discounting a spatially proximate low-preference POI. CONTEXTGOALPREDICTOR, which uses the same three encoder streams without explicit mental-state supervision, remains close to BDIBOTTLENECK on this metric, consistent with the hypothesis that preference signal leaks into the context representation even without direct desire supervision, but that explicit supervision sharpens the preference-proximity tradeoff at the distractor moment.

### F.5 Belief Head Sensitivity

For each (agent, POI) pair, we compute an evidence count from training episodes (how often that agent visited that POI) and a belief sensitivity score on test episodes, using the same $\Delta$ definition from Section 5.3.

Positive $\Delta$ indicates separation in the expected direction for that (agent, POI) pair. Because the belief head consumes only context encodings (not the current trajectory embedding $z_\tau$), this analysis is reported once rather than per path-fraction.

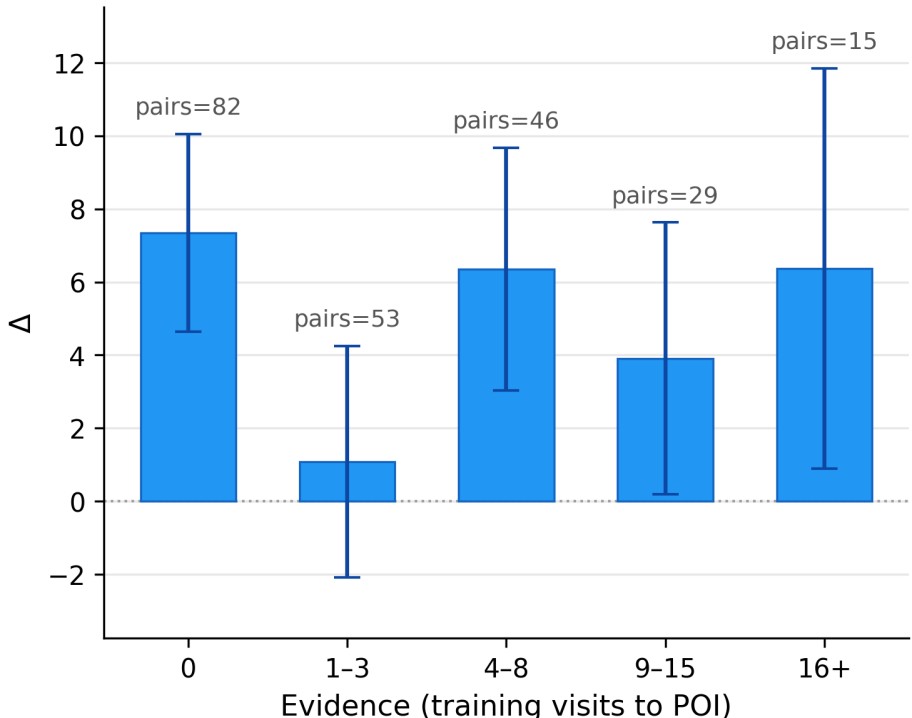

Figure 8: Belief-head sensitivity by training evidence on the test split. Bars show mean $\Delta$ per evidence bin, where $\Delta = \hat{P}(C \mid b_j{=}0) - \hat{P}(C \mid b_j{=}1)$. Error bars denote $\pm 1$ standard error across (agent, POI) pairs, and labels show the number of pairs in each bin.

Figure 8 summarizes mean $\Delta$ by evidence bin (0, 1–3, 4–8, 9–15, 16+ visits), with error bars showing standard error across (agent, POI) pairs. Most bins are positive, indicating non-trivial belief discrimination overall. However, the pattern is not monotonic in evidence: low-evidence and high-evidence bins can both be positive, and the 1–3 bin is near zero with wide uncertainty. This indicates that while the model captures belief-relevant signal, increased visit-frequency evidence alone does not produce a clean increase in sensitivity in this evaluation.

### F.6   Agent-Clustered Sensitivity Analysis for Belief Counterfactuals

To assess robustness to within-agent dependence, we run an agent-clustered sensitivity analysis for belief counterfactual effects: episode-level deltas are averaged within each agent, then one-sided tests are applied to the 15 agent-level means per split. The directional pattern is preserved (positive alternative gain, negative goal drop), but most clustered tests are weaker than episode-level tests; the clearest retained effect is goal-drop on the held-out new-agent split.

Table 15: Agent-clustered sensitivity analysis for counterfactual belief effects at 50% path (rank $\leq 3$). For each agent, episode-level deltas are averaged to one value per outcome ($n_{\text{agents}} = 15$ per split), then one-sided tests are applied to agent-level means.

| Split | Outcome | Mean | 95% CI | Permutation $p$ | Sign test $p$ |
|---|---|---|---|---|---|
| Test | $\Delta_{\text{alt}}$ | 0.0054 | [-0.0009, 0.0125] | 0.0784 | 0.1509 |
| Test | $\Delta_{\text{goal}}$ | -0.0122 | [-0.0259, 0.0013] | 0.0521 | 0.0592 |
| Test (new agent) | $\Delta_{\text{alt}}$ | 0.0050 | [-0.0040, 0.0141] | 0.1634 | 0.5000 |
| Test (new agent) | $\Delta_{\text{goal}}$ | -0.0233 | [-0.0338, -0.0130] | 0.0006 | 0.0037 |

### F.7 Belief Counterfactual: Full Path-Fraction Sweep

Table 16 reports mean $\Delta_{\text{alt}}$ and $\Delta_{\text{goal}}$ across all four evaluated path fractions for both splits. On the held-in test split, goal-drop magnitude decreases monotonically from 25% to 75% reveal (from $-0.020$ to $-0.010$), consistent with trajectory evidence increasingly outweighing the belief intervention as more of the path is observed. The held-out new-agent split shows a flatter or slightly growing pattern, suggesting that trajectory embeddings carry less agent-specific information for unseen agents, leaving the belief channel more influential throughout. Figure 9 shows the corresponding category-level effects at 25% reveal.

Table 16: Mean $\Delta_{\text{alt}}$ and $\Delta_{\text{goal}}$ for belief counterfactual interventions at each path fraction (rank $\leq 3$, $n = 206$ test / $n = 237$ new-agent episodes).

| Path fraction | Test | | Test (new agent) | |
|---|---|---|---|---|
| | $\Delta_{\text{alt}}$ | $\Delta_{\text{goal}}$ | $\Delta_{\text{alt}}$ | $\Delta_{\text{goal}}$ |
| 25% | +0.0120 | −0.0200 | +0.0013 | −0.0229 |
| 50% | +0.0061 | −0.0147 | +0.0055 | −0.0240 |
| 75% | +0.0040 | −0.0100 | +0.0017 | −0.0264 |
| 100% | +0.0072 | −0.0171 | +0.0007 | −0.0242 |

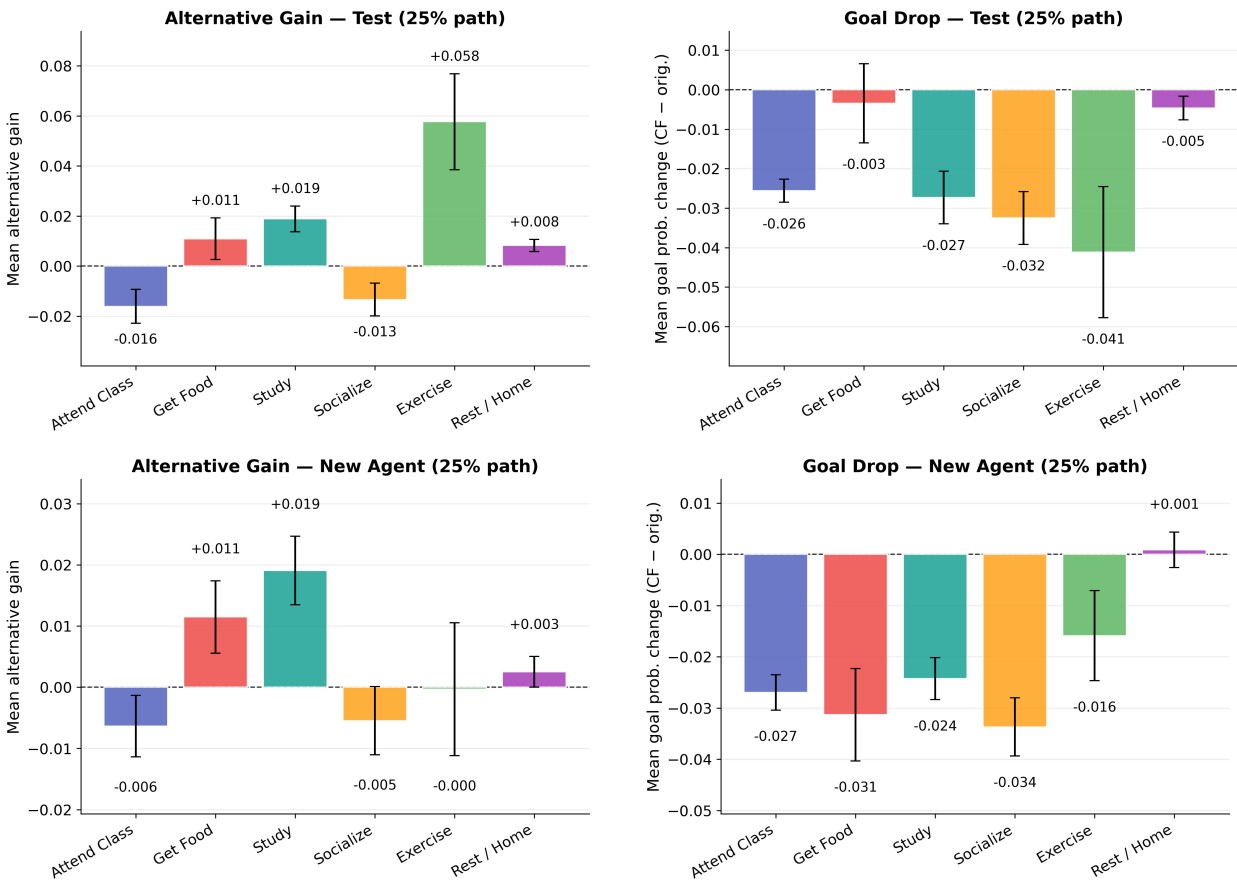

Figure 9: Belief counterfactual at 25% path reveal (rank $\leq$ 3, 2×2 panel). Top row: held-in test split; bottom row: held-out new-agent split. Left column: mean alternative gain. Right column: mean goal drop. Error bars: ±1 SE. Compare to the 50% results in Figure 4: goal-drop magnitude is larger here on held-in agents, consistent with weaker trajectory evidence at earlier reveals.

### F.8 Trajectory-Level Counterfactual Diagnostics

The local desire and belief interventions in the main text keep the observed trajectory prefix fixed and edit only the intermediate channel before the goal head. That is the right test for whether the representation is interventionable, but it is not a full behavioral counterfactual. We therefore ran two trajectory-level diagnostics to make this limitation explicit and to test the harder setting under controlled assumptions.

In the simulator-resampled diagnostic, we edit the simulator's mental state—closing the goal's belief entry (and opening same-category alternatives) for belief edits, or shifting to the next belief-feasible desire category for desire edits—and resample a counterfactual goal $g_{\text{sim}}$ from the edited feasible set. The simulator then regenerates a full near-shortest-path trajectory from the agent's original start toward $g_{\text{sim}}$, producing a fresh counterfactual episode rather than a suffix spliced onto the original prefix. The frozen model performs ordinary inference—with no intervention on its own channels—on a 50% prefix of this regenerated trajectory and is scored against $g_{\text{sim}}$. Table 17 reports top-1 accuracy, top-5 accuracy, and the probability the model assigns to $g_{\text{sim}}$. This diagnostic therefore asks a behavioral question—whether the model reads off the new goal from behavior the agent would actually produce under the edited mental state—and complements the model-planned diagnostic, which instead edits the model's own channels at a fixed prefix. Top-5 accuracy is substantially above top-1 because several POIs in the edited category typically remain plausible at the midpoint.

Table 17: Simulator-resampled full-trajectory counterfactual prediction at 50% path reveal. Values are means with 95% bootstrap confidence intervals clustered by agent. Top-1/top-5 measure agreement with the simulator-sampled counterfactual goal; $p(g_{\text{sim}})$ is the mean probability assigned to that goal.

| Regime | Split | Type | $n$ | Top-1 (%) | Top-5 (%) | $p(g_{\text{sim}})$ |
|---|---|---|---|---|---|---|
| dense | test | belief | 960 | 14.7 [11.2, 18.4] | 57.0 [50.0, 63.6] | 0.0902 [0.0812, 0.0989] |
| dense | test | desire | 960 | 14.0 [11.1, 17.1] | 50.9 [45.4, 56.8] | 0.0874 [0.0823, 0.0931] |
| dense | test_new_agent | belief | 960 | 11.6 [9.1, 14.3] | 50.9 [45.9, 55.7] | 0.0807 [0.0756, 0.0862] |
| dense | test_new_agent | desire | 960 | 14.6 [12.6, 16.7] | 55.8 [51.6, 60.2] | 0.0946 [0.0884, 0.1016] |
| sparse | test | belief | 480 | 17.5 [13.3, 21.9] | 58.8 [52.9, 64.6] | 0.0859 [0.0785, 0.0935] |
| sparse | test | desire | 480 | 9.6 [6.5, 12.9] | 45.6 [39.6, 52.5] | 0.0729 [0.0675, 0.0789] |
| sparse | test_new_agent | belief | 480 | 14.0 [9.2, 19.6] | 55.8 [49.8, 62.5] | 0.0767 [0.0707, 0.0830] |
| sparse | test_new_agent | desire | 480 | 9.6 [6.7, 12.9] | 47.9 [42.7, 53.3] | 0.0723 [0.0703, 0.0743] |

Table 18: Model-planned trajectory counterfactual probability deltas at 50% path reveal. Values are means with 95% bootstrap confidence intervals clustered by seed-agent. $\Delta_{\text{orig}} = p^{\text{cf}}(g_{\text{orig}}) - p(g_{\text{orig}})$; $\Delta_{\text{sel}} = p^{\text{cf}}(g_{\text{cf}}) - p(g_{\text{orig}})$; $\Delta_{\text{sim}} = p^{\text{cf}}(g_{\text{sim}}) - p(g_{\text{orig}})$; and $\Delta_{\text{sim}-\text{cf}} = p^{\text{cf}}(g_{\text{sim}}) - p^{\text{cf}}(g_{\text{cf}})$, where $g_{\text{cf}}$ is the model-selected counterfactual goal and $g_{\text{sim}}$ is the simulator-resampled counterfactual goal.

| Regime | Split | Type | Channel | $\Delta_{\text{orig}}$ | $\Delta_{\text{sel}}$ | $\Delta_{\text{sim}}$ | $\Delta_{\text{sim}-\text{cf}}$ |
|---|---|---|---|---|---|---|---|
| dense | test | belief | predicted | -0.0112 [-0.0137, -0.0088] | 0.0855 [0.0765, 0.0949] | -0.0870 [-0.0968, -0.0774] | -0.1725 [-0.1815, -0.1641] |
| dense | test | desire | predicted | -0.0705 [-0.0836, -0.0582] | 0.1220 [0.1042, 0.1391] | -0.0552 [-0.0719, -0.0399] | -0.1772 [-0.1888, -0.1652] |
| dense | test_new_agent | belief | predicted | -0.0095 [-0.0115, -0.0075] | 0.0976 [0.0883, 0.1074] | -0.0653 [-0.0736, -0.0576] | -0.1629 [-0.1719, -0.1543] |
| dense | test_new_agent | desire | predicted | -0.0532 [-0.0641, -0.0427] | 0.1467 [0.1275, 0.1647] | -0.0276 [-0.0420, -0.0137] | -0.1743 [-0.1880, -0.1602] |
| sparse | test | belief | predicted | 0.0016 [0.0009, 0.0022] | 0.0453 [0.0430, 0.0476] | -0.0234 [-0.0274, -0.0196] | -0.0687 [-0.0737, -0.0642] |
| sparse | test | desire | predicted | -0.0182 [-0.0223, -0.0148] | 0.0450 [0.0386, 0.0512] | -0.0265 [-0.0333, -0.0202] | -0.0715 [-0.0781, -0.0653] |
| sparse | test_new_agent | belief | predicted | 0.0010 [0.0004, 0.0016] | 0.0487 [0.0459, 0.0517] | -0.0216 [-0.0259, -0.0175] | -0.0703 [-0.0756, -0.0655] |
| sparse | test_new_agent | desire | predicted | -0.0140 [-0.0171, -0.0113] | 0.0484 [0.0430, 0.0537] | -0.0205 [-0.0253, -0.0157] | -0.0689 [-0.0757, -0.0626] |

In the model-planned diagnostic, the frozen model first predicts a counterfactual goal under the edited channel. We then route a new suffix from the intervention node to that model-selected goal, constructing a full model-imagined counterfactual path with the original prefix and a planned future. The paper-facing summary uses the same probability-delta style as the local interventions. $\Delta_{\text{orig}} = p^{\text{cf}}(g_{\text{orig}}) - p(g_{\text{orig}})$ measures whether the edited channel lowers probability on the original goal. $\Delta_{\text{sel}} = p^{\text{cf}}(g_{\text{cf}}) - p(g_{\text{orig}})$ measures the relative probability assigned to the model-selected counterfactual goal after intervention, using the original goal probability as the baseline. To connect the model-planned path to the simulator-resampled behavioral target, we also report two simulator-goal deltas. $\Delta_{\text{sim}} = p^{\text{cf}}(g_{\text{sim}}) - p(g_{\text{orig}})$ asks whether the edited model assigns the simulator counterfactual goal more probability than the original-goal baseline. $\Delta_{\text{sim}-\text{cf}} = p^{\text{cf}}(g_{\text{sim}}) - p^{\text{cf}}(g_{\text{cf}})$ directly compares the simulator counterfactual goal with the model-selected counterfactual goal under the same edited channel; values closer to zero mean the simulator goal is nearly as plausible as the model's selected goal, while more negative values mean the frozen model puts more probability on its own selected counterfactual goal. Table 18 shows that desire edits consistently produce negative $\Delta_{\text{orig}}$ and positive $\Delta_{\text{sel}}$ in both sparse and dense regimes. Belief edits are weaker: in dense data they produce the expected negative $\Delta_{\text{orig}}$, while in sparse data the model-planned belief delta at 50% is near zero and slightly positive. The new simulator-goal deltas separate two questions that were previously conflated: whether the intervention moves probability away from the original goal, and whether the model-selected counterfactual goal agrees in probability space with the simulator's counterfactual target. This matches the main conclusion that belief evidence is real but weaker than desire evidence, and that sparse false beliefs make the full trajectory-level belief setting especially hard.

Table 19 reports formal random-sign tests for the same displayed probability deltas. The test unit is the seed-agent cluster, so repeated episodes from the same simulated agent do not enter the permutation test as independent evidence. The expected direction is pre-specified for the two model-controlled quantities: $\Delta_{\text{orig}} < 0$ when the intervention moves probability away from the original goal, and $\Delta_{\text{sel}} > 0$ when the intervention raises probability on the model-selected counterfactual goal. The simulator-goal deltas are tested two-sided because either direction is scientifically informative: positive values mean the simulator goal gains probability relative to the baseline or approaches the model-selected goal, whereas negative values indicate remaining model-simulator disagreement.

Table 19: Cluster random-sign tests for the model-planned probability deltas in Table 18. Values are Holm-adjusted $p$-values across the displayed 50% predicted-channel tests. One-sided alternatives are used for the original-goal decrease and model-selected-goal increase; two-sided tests are used for simulator-goal deltas.

| Regime | Split | Type | $p_{\mathrm{adj}}(\Delta_{\mathrm{orig}} < 0)$ | $p_{\mathrm{adj}}(\Delta_{\mathrm{sel}} > 0)$ | $p_{\mathrm{adj}}(\Delta_{\mathrm{sim}} \neq 0)$ | $p_{\mathrm{adj}}(\Delta_{\mathrm{sim-cf}} \neq 0)$ |
|---|---|---|---|---|---|---|
| dense | test | belief | 0.0032 | 0.0032 | 0.0032 | 0.0032 |
| dense | test | desire | 0.0032 | 0.0032 | 0.0032 | 0.0032 |
| dense | test_new_agent | belief | 0.0032 | 0.0032 | 0.0032 | 0.0032 |
| dense | test_new_agent | desire | 0.0032 | 0.0032 | 0.0032 | 0.0032 |
| sparse | test | belief | 1.0000 | 0.0032 | 0.0032 | 0.0032 |
| sparse | test | desire | 0.0032 | 0.0032 | 0.0032 | 0.0032 |
| sparse | test_new_agent | belief | 1.0000 | 0.0032 | 0.0032 | 0.0032 |
| sparse | test_new_agent | desire | 0.0032 | 0.0032 | 0.0032 | 0.0032 |

The direct agreement view of this model-planned setting—how often the model's edited top goal matches the simulator-resampled counterfactual goal (top-1) or contains it in the edited top-5—is promoted to the main text (Table 4, Section 6.5), where it is easier to read as a behavioral counterfactual diagnostic than the probability-delta view here.

These trajectory diagnostics should be read as evidence about controlled counterfactual consistency, not as a claim that the model solves unrestricted behavioral counterfactual prediction. The simulator-resampled diagnostic provides a ground-truth behavioral target under an edited mental state; the model-planned diagnostic asks what future the frozen model itself would plan after the same kind of intervention. Both diagnostics use held-in test and held-out new-agent splits, and both are summarized with confidence intervals clustered by agent or seed-agent as appropriate.

## Appendix G  Top-1 Goal Inference Values (10% Fractions)

Table 20 reports the exact top-1 values underlying the main-text goal-inference curves at 10% path-fraction increments for the standard test and held-out new-agent splits, including the matched AUXCONTEXT baseline.

Table 20: Top-1 goal inference accuracy (%) at 10% path-fraction increments. Aux is the matched auxiliary-supervised context baseline.

| Path | Test | | | | Test (new agent) | | | |
|---|---|---|---|---|---|---|---|---|
| frac. (%) | Goal | Ctx | Aux | BDI | Goal | Ctx | Aux | BDI |
| 10 | 6.11 | 12.22 | 12.85 | 13.89 | 5.35 | 11.32 | 10.63 | 11.74 |
| 20 | 10.97 | 18.40 | 16.67 | 20.49 | 9.58 | 14.86 | 13.47 | 19.10 |
| 30 | 16.46 | 21.39 | 21.32 | 24.10 | 16.67 | 20.28 | 18.96 | 23.54 |
| 40 | 23.54 | 27.57 | 27.22 | 31.74 | 23.12 | 24.65 | 24.38 | 33.26 |
| 50 | 27.99 | 31.53 | 31.53 | 36.32 | 26.46 | 28.75 | 27.50 | 36.11 |
| 60 | 30.21 | 37.22 | 35.69 | 40.62 | 29.31 | 33.06 | 32.78 | 41.11 |
| 70 | 36.39 | 39.03 | 40.62 | 45.49 | 37.99 | 35.83 | 36.60 | 45.97 |
| 80 | 45.07 | 44.72 | 43.96 | 52.99 | 47.08 | 41.32 | 42.29 | 54.38 |
| 90 | 51.46 | 47.22 | 48.82 | 58.54 | 50.62 | 45.21 | 45.97 | 60.35 |
| 100 | 44.38 | 45.76 | 47.92 | 55.56 | 45.69 | 43.33 | 42.78 | 56.88 |

## Appendix H  Statistical Significance of Goal Inference Gains

Tables 21 and 22 report McNemar's exact paired tests comparing BDIBottleneck against each baseline at all ten 10%-increment path fractions, on both in-distribution and held-out new-agent splits. $p$-values are adjusted with Holm–Bonferroni correction across 40 simultaneous tests ($\alpha = 0.05$). BDIBottleneck significantly outperforms GoalPredictor at most fractions on both splits (21 of 40 tests significant overall). Gains over ContextGoalPredictor emerge most consistently at longer path reveals (80–100%) and on held-out agents (70–100%), consistent with the hypothesis that explicit mental-state supervision provides the greatest

Table 21: McNemar's exact test (BDIBottleneck vs. baselines) at 10% path-fraction increments. Holm–Bonferroni adjusted $p$-values across 40 tests. **Bold** = significant at $\alpha = 0.05$. Top panel: 10–50%; bottom panel: 60–100%.

| Split | Comparison | 10% | 20% | 30% | 40% | 50% |
|---|---|---|---|---|---|---|
| test | BDI vs. GP | **< 0.001** | **< 0.001** | 0.282 | **< 0.001** | **0.001** |
| | BDI vs. CTX | 1.000 | 1.000 | 1.000 | 1.000 | 1.000 |
| test_new_agent | BDI vs. GP | **0.001** | **< 0.001** | 0.276 | **< 0.001** | **< 0.001** |
| | BDI vs. CTX | 1.000 | 1.000 | 0.919 | 1.000 | 1.000 |

| Split | Comparison | 60% | 70% | 80% | 90% | 100% |
|---|---|---|---|---|---|---|
| test | BDI vs. GP | **< 0.001** | **< 0.001** | **0.005** | **0.045** | **< 0.001** |
| | BDI vs. CTX | 0.114 | 0.479 | **0.047** | **0.007** | 0.145 |
| test_new_agent | BDI vs. GP | **< 0.001** | 0.120 | 1.000 | 0.191 | **0.002** |
| | BDI vs. CTX | 1.000 | **0.003** | **0.004** | **0.001** | **0.005** |

Table 22: Full McNemar's exact test results at 10% path-fraction increments. $n_{01}$: BDI correct & baseline wrong; $n_{10}$: baseline correct & BDI wrong; $p$: raw two-sided $p$-value; $p^*$: Holm–Bonferroni adjusted (40 tests total). **Bold** = significant at $\alpha = 0.05$.

| Split | Comparison | Frac | $n$ | $n_{01}$ | $n_{10}$ | $p$ | $p^*$ |
|---|---|---|---|---|---|---|---|
| test | BDI vs. GoalPredictor | 10% | 480 | 15 | 51 | < 0.001 | **< 0.001** |
| | | 20% | 480 | 19 | 69 | < 0.001 | **< 0.001** |
| | | 30% | 480 | 44 | 70 | 0.019 | 0.282 |
| | | 40% | 480 | 39 | 90 | < 0.001 | **< 0.001** |
| | | 50% | 480 | 40 | 86 | < 0.001 | **0.001** |
| | | 60% | 480 | 36 | 97 | < 0.001 | **< 0.001** |
| | | 70% | 480 | 39 | 91 | < 0.001 | **< 0.001** |
| | | 80% | 480 | 43 | 86 | < 0.001 | **0.005** |
| | | 90% | 480 | 45 | 80 | 0.002 | **0.045** |
| | | 100% | 480 | 40 | 91 | < 0.001 | **< 0.001** |
| | BDI vs. ContextGoalPredictor | 10% | 480 | 34 | 37 | 0.813 | 1.000 |
| | | 20% | 480 | 41 | 47 | 0.594 | 1.000 |
| | | 30% | 480 | 45 | 56 | 0.320 | 1.000 |
| | | 40% | 480 | 47 | 62 | 0.180 | 1.000 |
| | | 50% | 480 | 50 | 63 | 0.259 | 1.000 |
| | | 60% | 480 | 42 | 72 | 0.006 | 0.114 |
| | | 70% | 480 | 49 | 73 | 0.037 | 0.479 |
| | | 80% | 480 | 45 | 80 | 0.002 | **0.047** |
| | | 90% | 480 | 42 | 83 | < 0.001 | **0.007** |
| | | 100% | 480 | 50 | 81 | 0.009 | 0.145 |
| test_new_agent | BDI vs. GoalPredictor | 10% | 480 | 11 | 41 | < 0.001 | **0.001** |
| | | 20% | 480 | 16 | 68 | < 0.001 | **< 0.001** |
| | | 30% | 480 | 37 | 61 | 0.020 | 0.276 |
| | | 40% | 480 | 31 | 77 | < 0.001 | **< 0.001** |
| | | 50% | 480 | 32 | 81 | < 0.001 | **< 0.001** |
| | | 60% | 480 | 31 | 83 | < 0.001 | **< 0.001** |
| | | 70% | 480 | 49 | 81 | 0.006 | 0.120 |
| | | 80% | 480 | 60 | 80 | 0.108 | 1.000 |
| | | 90% | 480 | 52 | 82 | 0.012 | 0.191 |
| | | 100% | 480 | 40 | 85 | < 0.001 | **0.002** |
| | BDI vs. ContextGoalPredictor | 10% | 480 | 42 | 40 | 0.912 | 1.000 |
| | | 20% | 480 | 44 | 59 | 0.167 | 1.000 |
| | | 30% | 480 | 47 | 49 | 0.919 | 0.919 |
| | | 40% | 480 | 55 | 64 | 0.463 | 1.000 |
| | | 50% | 480 | 56 | 71 | 0.214 | 1.000 |
| | | 60% | 480 | 58 | 74 | 0.191 | 1.000 |
| | | 70% | 480 | 44 | 89 | < 0.001 | **0.003** |
| | | 80% | 480 | 49 | 95 | < 0.001 | **0.004** |
| | | 90% | 480 | 44 | 92 | < 0.001 | **0.001** |
| | | 100% | 480 | 42 | 84 | < 0.001 | **0.005** |

benefit when generalising to unseen agents. The within-agent clustering structure (15 agents $\times \approx 32$ episodes each) means that McNemar's exchangeability assumption is mildly violated; treating episodes as independent therefore slightly underestimates the true standard error, and borderline $p$-values should be interpreted conservatively.

# Appendix I   Top-5 Goal Inference

Top-5 goal accuracy curves are provided here as a complement to the top-1 analysis in Section 6. At top-5, the original three learned models (GoalPredictor, ContextGoalPredictor, and BDIBottleneck) improve substantially, narrowing the relative gap between them. BDIBottleneck continues to lead on both splits across all path fractions, and the separation between BDIBottleneck and ContextGoalPredictor remains visible, particularly at mid-range fractions (50–75%) where the desire bottleneck provides the most disambiguation benefit. The near-convergence of all models at 100% path reveal on the new-agent split indicates that full trajectory information partially compensates for the lack of mental-state supervision when the task is relaxed from top-1 to top-5.

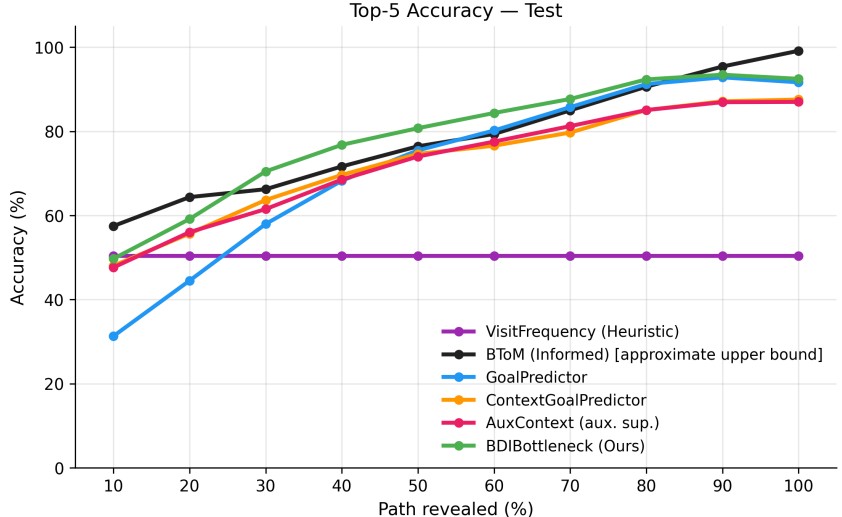

Figure 10: Top-5 goal accuracy vs. path fraction on the held-in test split for the three learned models.

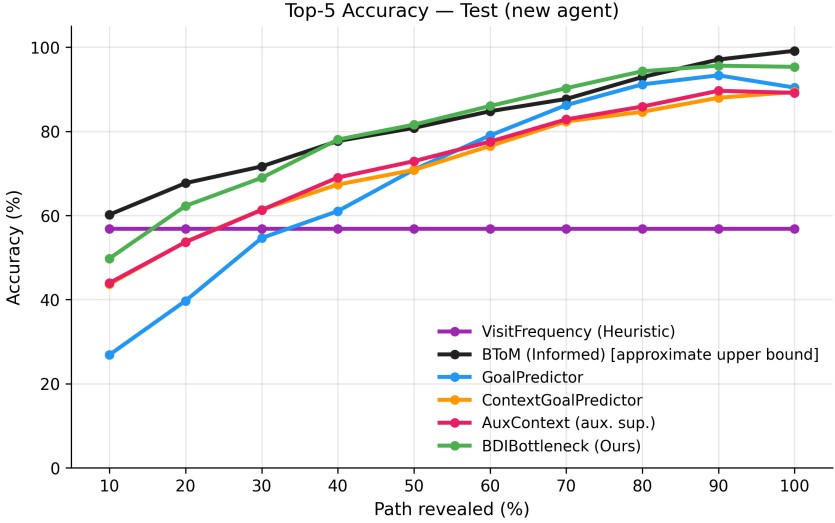

Figure 11: Top-5 goal accuracy vs. path fraction on the held-out new-agent split. The generalization gap between test and new-agent performance narrows relative to the top-1 setting, consistent with the relaxed identification requirement.

## Appendix J    Desire Counterfactual: Full Path-Fraction Sweeps

The main results section reports desire counterfactual delta matrices at 50% path reveal for both prototype modes on both evaluation splits. This appendix provides the complete four-fraction sweep (25%, 50%, 75%, 100%) for both prototype modes and both splits. The sweep shows that the off-diagonal structure — substituting category $j$ increases predicted probability mass on POIs in category $j$ — is present at every fraction and strengthens as path reveal increases. This progression is expected: a longer observed prefix provides more trajectory-based disambiguation, so the goal head responds more sharply to the desire signal at higher fractions. At 25% reveal the delta values are smaller in magnitude but the directional pattern is consistent, indicating that desire substitution influences goal prediction even under severe partial observability. The off-diagonal significance reported in Table 2 holds at all four fractions (permutation $p \leq 10^{-4}$ in every case); 50% is reported as the representative fraction for the main text.

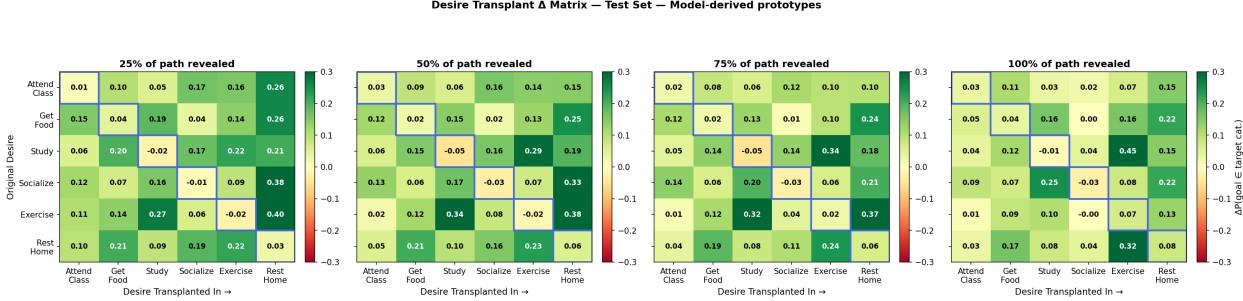

Figure 12: Full four-fraction desire counterfactual delta matrices, model-derived prototypes, held-in test split. Panels correspond to 25%, 50%, 75%, and 100% path reveal (left to right). Rows are source desire categories; columns are transplanted categories; each cell is the mean change in category-level goal probability relative to the no-swap baseline.

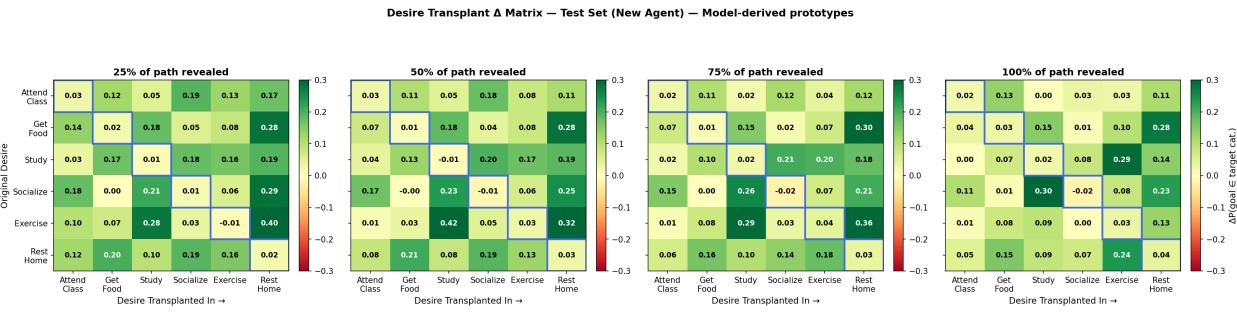

Figure 13: Full four-fraction desire counterfactual delta matrices, model-derived prototypes, held-out new-agent split. The off-diagonal structure generalizes across held-out agents, indicating that the desire channel's influence on goal prediction is not agent-specific.

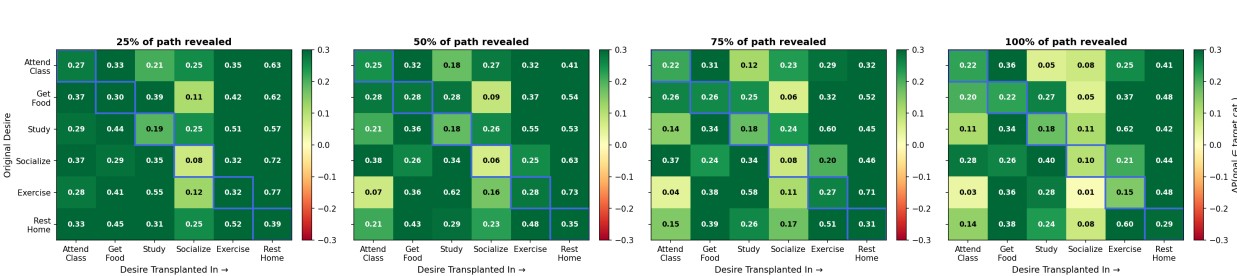

Figure 14: Full four-fraction desire counterfactual delta matrices, handcrafted category-only prototypes, held-in test split. Handcrafted prototypes place uniform mass over the four POIs in the target category and zero elsewhere, producing a sharper desire signal than model-derived prototypes and correspondingly larger delta magnitudes.

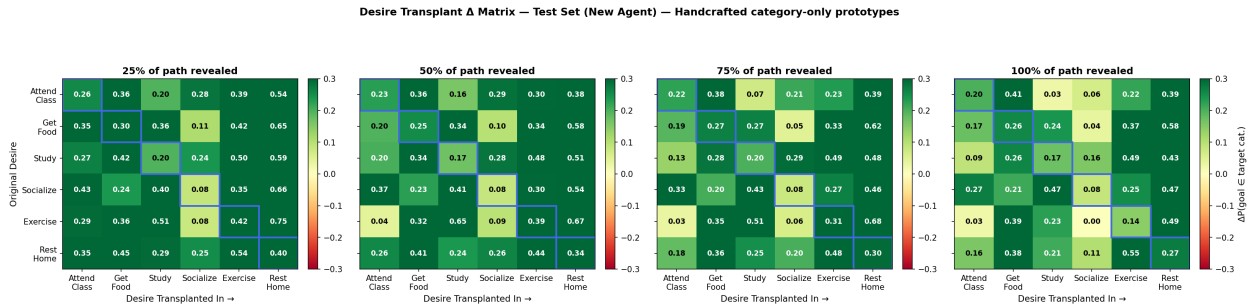

Figure 15: Full four-fraction desire counterfactual delta matrices, handcrafted category-only prototypes, held-out new-agent split. The stronger category signal of handcrafted prototypes produces consistently larger shifts than model-derived prototypes across both splits and all fractions.

## Appendix K   Desire Prototype Distributions

Figures 16–19 visualize the prototype vectors used in the desire counterfactual experiments. Model-derived prototypes are computed as the mean of all inferred desire vectors $\hat{\pi}_i$ within each category, aggregated over evaluation episodes. Because individual desire estimates are smooth distributions over all 24 POIs rather than one-hot category indicators, model-derived prototypes retain non-trivial mass outside the target category. This spread explains why model-derived prototypes produce smaller delta magnitudes than handcrafted prototypes: the transplanted signal is less categorically concentrated. Handcrafted prototypes are deterministic and category-specific by construction, serving as an idealized upper bound on the sharpness of the desire signal.

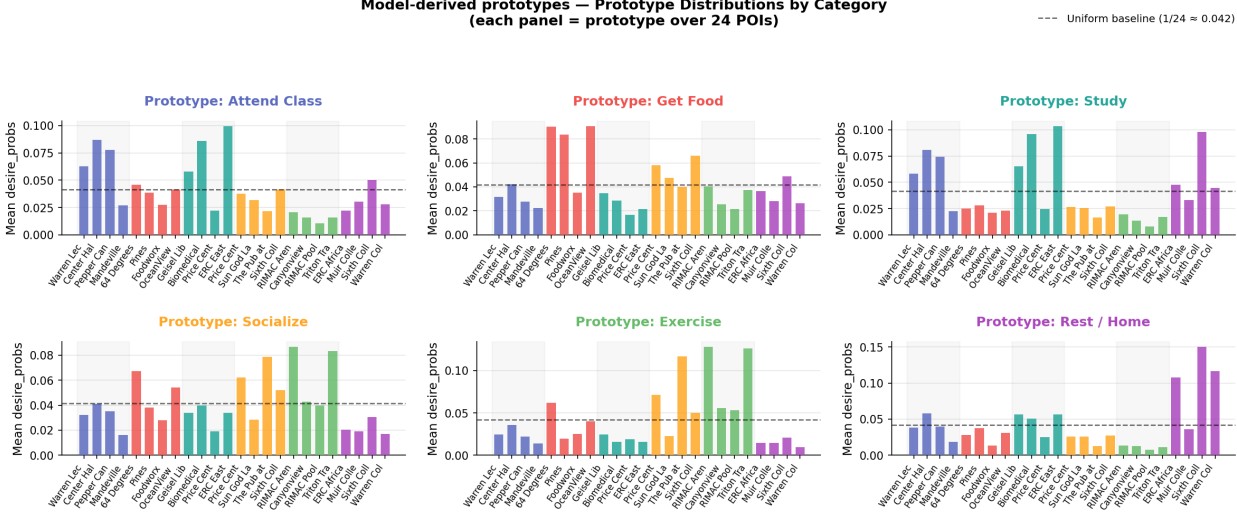

Figure 16: Model-derived prototype distributions on the held-in test split. Each panel shows the 24-POI mean desire vector for one target category, with POIs grouped and color-coded by category. Non-trivial mass outside the target category reflects the smoothness of individual desire estimates.

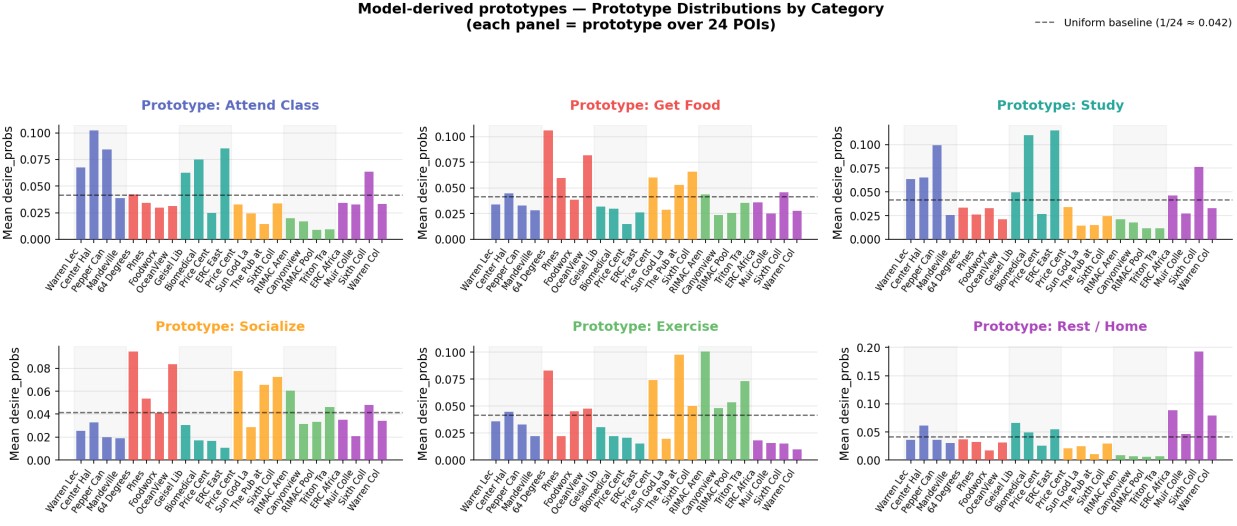

Figure 17: Model-derived prototype distributions on the held-out new-agent split. The within-category concentration is comparable to the test split, indicating that the desire encoder learns category-consistent representations that generalize to unseen agents.

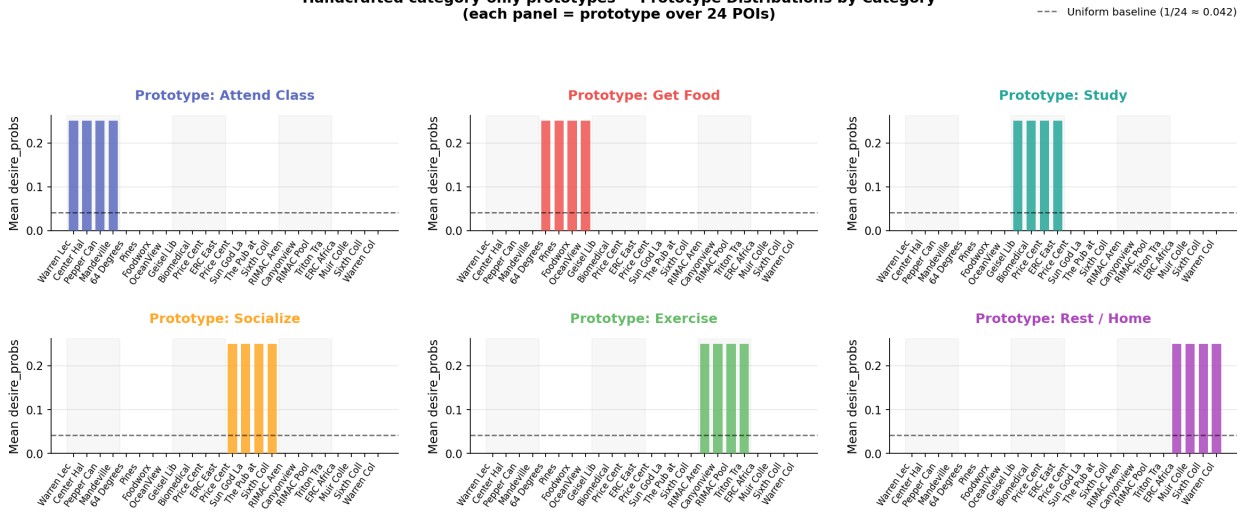

Figure 18: Handcrafted category-only prototype distributions on the held-in test split. Each prototype assigns uniform mass over the four POIs in the target category and zero mass elsewhere, providing a sharp, idealized desire signal for comparison with model-derived prototypes.

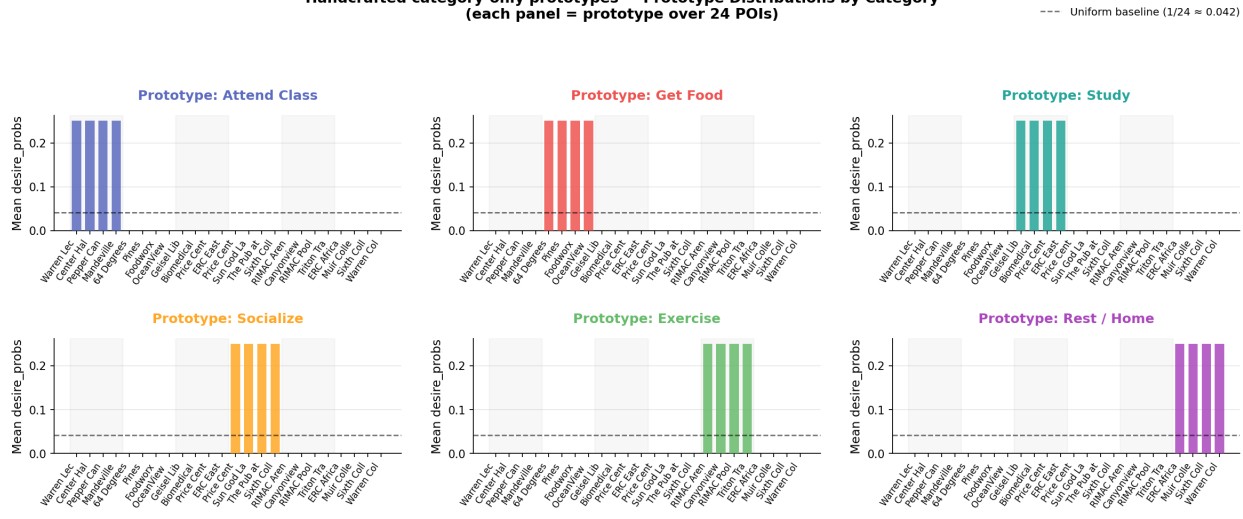

Figure 19: Handcrafted category-only prototype distributions on the held-out new-agent split. Identical in construction to Figure 18; shown separately to confirm that prototype construction is split-invariant by design.

