# OpenReview forum: "Structured Machine Theory of Mind from Agent Trajectories"
_TMLR — Under review for TMLR_

### Review · Reviewer_YsSq · 2026-06-19

**Summary Of Contributions:**

The paper introduces Structured Machine Theory of Mind (SMToM), instantiated as the BDIBottleneck architecture. It infers explicit belief and desire representations from agent trajectories and routes them through a Belief-Desire-Intention (BDI) bottleneck to predict goals. This structure allows the model to not only perform accurate goal inference but also natively support counterfactual interventions (e.g., asking how behavior changes if the agent's beliefs or desires are altered).

Strengths:
1. Elegant architectural design that bridges the interpretability of model-based ToM with the scalability of learning-based ToM.
2. The counterfactual intervention experiments are well-designed and convincingly demonstrate that the learned representations are causally active rather than merely correlational.
3. Thorough evaluation on both in-distribution and held-out agent splits.

Weaknesses:
1. The framework relies heavily on explicit supervision of ground-truth beliefs and desires, which are available in this controlled simulator but absent in real-world datasets.
2. The simulated environment is relatively simple (a static graph with 24 POIs), and the belief configurations are sparse (only 0-5 closed POIs per episode).

**Audience:**

Yes

**Audience Explanation:**

The paper sits at the intersection of trajectory prediction, cognitive modeling, and interpretable machine learning. Researchers working on autonomous agents, human-robot interaction, and causal representation learning will find the BDI bottleneck approach and the counterfactual evaluation methodology highly relevant and interesting.

**Broader Impact Concerns:**

While the current work is confined to a simulated pedestrian domain, the ultimate goal of inferring latent human mental states (beliefs and desires) from mobility traces raises significant privacy and surveillance concerns. If deployed in real-world tracking systems, such models could be used to infer sensitive personal preferences without consent. The authors should add a brief Broader Impact Statement acknowledging these dual-use risks and privacy implications for future real-world applications.

**Claims And Evidence:**

Yes

**Claims Explanation:**

The empirical evidence strongly supports the claims. The authors provide rigorous ablation studies (NoDesire, NoBelief) that isolate the contributions of the BDI components. Furthermore, the counterfactual evaluations (substituting desire prototypes and altering belief states) show statistically significant shifts in goal predictions in the expected causal directions. The use of an approximate upper bound (BToM Informed) also helps ground the performance metrics effectively.

**Requested Changes:**

Critical to securing acceptance:

Add a dedicated section or expand the discussion detailing concrete strategies for applying this framework to real-world datasets where ground-truth belief and desire labels are unavailable. What specific proxy variables or unsupervised discovery methods could bridge this gap?

Would strengthen the work:
1. Include a brief discussion on how the BDIBottleneck would scale to more complex environments (e.g., continuous state spaces, dynamic environments requiring sequential belief updates, or larger graphs).
2. The authors note that the sparse belief configurations (0-5 closed POIs) limit the signal. It would strengthen the paper to include a minor synthetic experiment or robustness check with denser false-belief scenarios to demonstrate how the model handles more complex belief states.

---

### Review · Reviewer_1nKR · 2026-07-02

**Summary Of Contributions:**

The paper proposes Structured Machine Theory of Mind and a BDIBottleneck model for predicting agent goals from trajectories. The model learns explicit desire and belief channels from same-agent context and feeds them, along with the current trajectory embedding, into a goal predictor. This design allows direct interventions on predicted desires/beliefs and tests whether goal predictions change in the expected direction.

The main strengths are the clear motivation, interpretable architecture, controlled simulator with mental-state labels. However, the paper has important weaknesses. The setting is synthetic and uses oracle supervision for beliefs/desires. The belief intervention effects are small and partly fragile, and the counterfactual tests are local rather than full trajectory counterfactuals.

**Audience:**

Yes

**Audience Explanation:**

Some TMLR readers would likely be interested in this paper because it studies an important problem: how to make trajectory models more interpretable and more suitable for mental-state or counterfactual reasoning. The idea of putting explicit belief/desire variables into a neural trajectory model is relevant to researchers in machine theory of mind, interpretable representation learning, causality learning, etc.

However, the potential audience interest can be limited by the synthetic setup, the need for ground-truth mental-state labels, and the weak belief intervention results.

**Broader Impact Concerns:**

The paper studies inference of beliefs, preferences, and goals from trajectory data. If applied to real human mobility data, this could raise privacy and surveillance concerns. In the current submission, the experiments are synthetic, so immediate harm is limited.

**Claims And Evidence:**

No

**Claims Explanation:**

The paper provides some useful evidence for a weaker claim, adding explicit belief/desire supervision can improve goal prediction, and desire interventions produce clear shifts in predicted goal categories. However, the stronger claims about learning a causally meaningful BDI representation is not fully supported.

Firstly, the main neural baseline is confounded. BDIBottleneck is compared with ContextGoalPredictor, but BDIBottleneck has extra auxiliary supervision for desire and belief, while ContextGoalPredictor does not. So it seems to me, the experiments do not separate the effect of the BDI bottleneck from the effect of extra supervised multi-task training very clearly. A fairer baseline would use the same auxiliary losses but remove the bottleneck.

Also, the belief counterfactual evidence is weak. The goal-probability drop after closing the believed goal is small, -0.0147 on the test set and -0.0240 on the new-agent set. Also, when the authors cluster by agent, the held-in test result becomes only borderline significant (0.052).

Moreover, the counterfactual experiments are not full behavioral counterfactuals. The paper keeps the observed trajectory prefix fixed and only changes the intermediate belief/desire channel before the goal head. This tests only **local** model controllability, but it does not show that the model can correctly predict what the agent would actually do under a changed belief or desire.

The setup is useful for a controlled proof of concept, but it avoids the harder real-world setting where such labels are unavailable. The authors also acknowledges this limitation (external validity to real mobility behavior remains open).

**Requested Changes:**

1. Compare BDIBottleneck against an unbottlenecked model trained with the same desire and belief auxiliary losses. This is needed to separate the effect of the BDI bottleneck from the effect of extra multi-task supervision.
2. The paper should state that the counterfactual experiments test local interventions, not full counterfactual behavior under changed beliefs/desires.
3. I recommend the authors to strengthen belief-intervention evidence/results and avoid overstating significance.
4. False-belief F1 is low despite full supervision in a clean simulator. The authors should analyze why the belief module is weak and whether the belief channel is actually used by the goal predictor.
5. If possible, generate true counterfactual rollouts under changed beliefs/desires and compare model predictions to those outcomes.

---

### Review · Reviewer_M56P · 2026-07-14

**Summary Of Contributions:**

**Summary**

The authors motivate their work by arguing that current predictive human behavior models directly optimize for observed accuracy rather than deploying latent theory of mind reasoning. As a result, the authors question the ability of those models for making predictions under altered or counterfactual believe states and propose a 'Structured Machine Theory of Mind' (SMToM), that explicitly models belief and desire states to allow for latent interventions and counterfactual evaluations.

The approach is tested on a synthetic pedestrian navigation task and benchmarked against trajectory-only and context-aware baselines. The authors claim that the proposed SMToM method reliably identifies agent goals during the early to mid stages of path sequences. The authors furthermore consider counterfactual navigation sequences where agent internals goals are swapped, leading to a reliable shift in goal prediction; and, finally, marking locations as unavailable reduces prediction probability. The overall presented results seem to indicate a successful application of the method for the tested setting.



**Strengths**

1. The authors discuss an important aspect of latent ToM modeling when making predictions for 'intelligent' agents' actions. The author motive the need for the proposed ToM modeling well, and experimentally demonstrate the shortcomings of methods lacking this component.
2. The paper is generally well written and follows a common thread. Figure one clearly maps the architecture and loss terms and following tables and figures visualize and support the obtained results well.
3. Although tested on a rather confined environment, the authors thoroughly evaluate their approach and seem to obtain reasonable to good performance, with the compared baselines clearly lacking behind. Particularly, experiments applying interventional and counterfactual changes support the initial discussion and need for ToM models.





**Weaknesses**

Coming from a 'Pearlian' causal and AI/ML background, I found the paper to focus mostly on implementation details after an initially very exiting motivation. Opinions on what constitutes a 'structured' ToM might diverge depending on the reader background. Given the claims on structureal ToM, I expected a more formal analysis on the assumed (causal) structure that was assumed to be modeled or recovered by the proposed architecture, which the paper seems to lack; beyond introducing a rather general (although practically effective) belief/desire split in the architecture. Given that the rest of the paper was setup well and I strongly promote the overall presented idea, I would like to use this part to more broadly discuss the overall framing and the theoretical aspects of the paper, rather than pointing out any specific weaknesses:



**Lack of Theoretical Discussion.** While the paper is well motivated and seems to successfully tackle the task of behavior prediction from the promising direction of structured causal ToM, the presented work lacks any real discussion on explicit ToM structures or formal modeling of the assumed causal structure (in the sense of mentioning structural [neuro] causal models, graph NNs, Bayesian networks, any other graph-explicit structure; modeling the scenario via a causal model, or including any explicit causal inference step as part of the pipeline) as could be expected from the title or introduction. The only reference and theoretical foundation of the proposed DIBottleneck architecture seemingly to be an anonymous paper which is neither provided as supplementary material, nor any further discussed. There are no mentions on the assumed causal modeling of the overarching scenario or any provision of an identifiability proof (for example related to assumptions on the assumed scenario or the provided data). As such, theoretical aspects, relations to the mentioned hierarchical latent/causal structure identification or any other structured ToM theory remains are completely absent and can not be checked.



**Notion of Causality.** Throughout the paper, the authors often refer to the term 'causal'/'causally'. In conjunction with their proposal of a 'structural' ToM this should immediately links their work to the rather well established field of 'Pearlian' causality and structural causal models (SCM; [1]). In this theory, particular claims on latent structure identifiability (if not assumed a-priori) and can sometimes be downgraded to 'standard' latent identification. However, a thorough causal analysis and identifiability proof of latent factors, usually demands a causal model of the scenario and is governed under the do-calculus and the Pearlian Causal Hierarchy. A particular question that should be answered in this regard, would for example be: 'how do the observed actions inform the ToM state of the agent, and in turn influence their predictions and how does the proposed architecture reflect and model the assumed causal structure'. The exact definition of what constitutes 'causality' outside the particular Pearlian perspective (e.g. from the adjacent cognitive science domain) is a common point of discussion. While the authors might intended to more broadly refer to less strict notion of causality when using the terms 'causal', 'intervention' and 'counterfactuals', their particular choice of framing their approach as a structured theory, but lack of causal modeling, in conjunction with the absence of identifiability proofs, lead to a paper that makes it hard to draw any insights or guarantees that go beyond the presented experimental setup.



On this point, the authors seem to discuss the presented setting from a more general cognitive science and ML perspective. Considering the mentioned inclusion of causal aspects, papers on causal ToM/counterfactual [2], causality & cognitive science [3,4] or more general causal representation learning methods [5-9] come to mind.



**Generalization performance of BDIBottleneck Model.** Considering that the deployment of (causal or) structured approaches is commonly linked to seeking improved predictions robustness in O.O.D. settings by identifying the underlying causal relations or reasoning structure, it is rather odd that the proposed method seems to feature the worst generalization gap of all compared methods in figure 2. I would kindly like to ask the authors to explain this observation.



**Minor Points**

* The paper would be slightly better to read if the figures where included as PDF/vector graphics instead of rasterized images.
* The scaling of figures varies greatly within the paper. The authors might want to apply a more uniform scaling across figures.
* Margins between all tables and their captions seem to be visibly reduced. I'd like to recommend restoring this to default values.





[1] Pearl, Judea. *Causality*. Cambridge university press, 2009.
[2] Compagno, Dario, and Fabio Massimo Zennaro. "Teleological Inference in Structural Causal Models via Intentional Interventions." *arXiv preprint arXiv:2603.18968* (2026).
[3] Jin, Emily, et al. "MARPLE: A benchmark for long-horizon inference." *Advances in Neural Information Processing Systems* 37 (2024): 108824-108850.
[4] Wu, Sarah, Shruti Sridhar, and Tobias Gerstenberg. "That was close! a  counterfactual simulation model of causal judgments about decisions." *Proceedings of the annual meeting of the cognitive science society*. Vol. 44. No. 44. 2022.
[5] Li, Zijian, et al. "On the identification of temporal causal representation with instantaneous dependence." *International Conference on Learning Representations*. Vol. 2025. 2025.
[6] Yang, Mengyue, et al. "Causalvae: Disentangled representation learning via neural structural causal models." *Proceedings of the IEEE/CVF conference on computer vision and pattern recognition*. 2021.
[7] Javaloy, Adrián, Pablo Sánchez-Martín, and Isabel Valera. "Causal normalizing flows: from theory to practice." *Advances in Neural Information Processing Systems* 36 (2023): 58833-58864.
[8] Xia, Kevin, et al. "The causal-neural connection: Expressiveness, learnability, and inference." *Advances in Neural Information Processing Systems* 34 (2021): 10823-10836.
[9] Almodóvar, Alejandro, et al. "DeCaFlow: A deconfounding causal generative model." *Advances in Neural Information Processing Systems* 38 (2026): 136612-136666.

**Audience:**

No

**Audience Explanation:**

The paper is well motivated and presents an interesting structured ToM approach. As already discussed in the prior point, the lack of theoretical analysis does not allow any judgment on the formal validity of the proposed method.

**Broader Impact Concerns:**

None.

**Claims And Evidence:**

No

**Claims Explanation:**

While the presented pipeline seems to be soundly setup and experiments well conducted and interpreted. The paper fully lacks a formal modeling and theoretical analysis on the structured theory of mind as motivated in the title and introduction. It is unclear therefore unclear whether, and under which conditions, results are expected to generalize.

**Requested Changes:**

As already mentioned in the section above, I generally believe in the proposed idea, but see the lack of causal modeling and general theoretical analysis as the major concern that needs to be addressed by the authors. In its current state, the paper solely relies on an unknown and unprovided workshop paper. Identifiability proofs should be repeated in the paper and applicability to the setting of this paper needs to be shown.

Generally, the authors should be more explicit how their architecture leverages or implements the claimed structured ToM. Particularly, clarifying the relation between this work, the exact relation to more formal Pearlian causal SCMs and possibly causal representation identifiability might provide the needed theoretical insights.

---

> ### Comment · Reviewer_M56P · 2026-07-16
>
> Dear authors,
> as all reviews are now also visible to us reviewers (and seem to slightly diverge in their judgment), I would like to make the following comments/corrections:
>
> **Basis of Theoretical Discussion.** Upon reconsidering the particular paragraph of the paper again, I noticed that I misread the sentence of the paper, which claims that the author's approach is actually *distinct* from the cited anonymous paper. I, hereby, want acknowledge and apologize for this mistake. Please disregard my comments on this point in the original review. However, my concerns on the absence of a theoretical analysis/discussion on assumptions remain.
>
> **Findings of this Paper.** Considering the interpretation of the question on 'the interest of the TMLR community in the findings of the paper', I decided to select 'no' as a result of the (possible lack of) theoretical discussion, carrying on from the previous 'claims made' question. I, otherwise, share the sentiment of reviewer 1nKR, considering a slightly restrictive synthetic setup, and I am confident that findings will attract interest with in the TMLR community if the authors are able to provide a brief formal analysis/discussion on assumptions for their method.
>
>
>
> **Clarification of Feedback.** Given that my review discusses a (possibly) broader shortcoming of the paper, while the other reviews seem to consider rather particular methodological and experimental weaknesses, I'd like to clarify that I do not intend to challenge the fundamental presentation or the existing contributions of the paper in its current form. Rather, I hope that these comments will be useful to further strengthen it's overall soundness and contribution.